# Catastrophic slab loss in southwestern Pangea preserved in the mantle and igneous record

Guido M. Gianni [1,2,3 ✉] & César R. Navarrete [1,2]

The Choiyoi Magmatic Province represents a major episode of silicic magmatism in southwestern Pangea in the mid-Permian-Triassic, the origin of which remains intensely debated. Here, we integrate plate-kinematic reconstructions and the lower mantle slab record beneath southwestern Pangea that provide clues on late Paleozoic-Mesozoic subducting slab configurations. Also, we compile geochronological information and analyze geochemical data using tectono-magmatic discrimination diagrams. We demonstrate that this magmatic event resulted from a large-scale slab loss. This is supported by a paleogeographic coincidence between a reconstructed 2,800-3,000-km-wide slab gap and the Choiyoi Magmatic Province and geochemical data indicating a slab break-off fingerprint in the latter. The slab break-off event is compatible with Permian paleogeographic modifications in southwestern Pangea. These findings render the Choiyoi Magmatic Province the oldest example of a geophysically constrained slab loss event and open new avenues to assess the geodynamic setting of silicic large igneous provinces back to the late Paleozoic.

---

[1] Consejo Nacional de Investigaciones Científicas y Técnicas (CONICET), Capital Federal, Argentina. [2] Laboratorio Patagónico de Petro-Tectónica. Universidad Nacional de la Patagonia "San Juan Bosco", Dpto. de Geología, F.C.N, Comodoro Rivadavia, Chubut, Argentina. [3] Instituto Geofísico Sismológico Ing. Fernando Volponi (IGSV), Universidad Nacional de San Juan, San Juan, Argentina. ✉email: guidogianni22@gmail.com

After more than 30 years of scientific debate, the geodynamic setting of the Choiyoi Magmatic Province[1], one of the largest silicic magmatic events of Pangea, is still not understood (Fig. 1a, b). This magmatic province is composed of mesosilicic and silicic rocks of Lower Permian to Lower Triassic age that record a compositional change from calc-alkaline/transitional in the lower section to alkaline in the upper section[2,3]. Due to the areal extent (909,250 km$^2$), volume (947,553 km$^3$), and time span [ca. 40 million years (Myr)], the Choiyoi Magmatic Province was classified as a silicic large igneous province (SLIP)[1,3,4]. This SLIP was emplaced between ca. 286 and 247 Ma (million years ago)[2], with local expressions as early as ca. 290-295 Ma, and records a magmatic flare-up stage from ca. 275–250 Ma[3]. The magmatic activity took place in an overall extensional to transtensional regime along the southwestern Pangea margin[4–7] (Fig. 1a). However, the magmatism was locally punctuated by shortening events in the southernmost Andean and North Patagonian Massif regions of this magmatic province (e.g., [8,9]) (Fig. 1b). Understanding the origin of the Choiyoi Magmatic Province has global implications as this SLIP might have contributed to the Permian-Triassic mass extinction[5,10,11], one of the most severe ecosystems collapses of the Phanerozoic[12].

There are two end-member hypotheses proposed to explain the development of the Choiyoi Magmatic Province. One suggests an origin associated with a very slow convergence or halted subduction scenario produced by slab break-off episodes[1,13–18] [i.e., a detachment of the subducting plate caused by horizontal slab tearing[19]]. The other hypothesis envisions a continued subduction scenario[3,20–25] associated with changes in slab dip[5,24–26]. Other studies have included components from both proposals suggesting a convergent setting at ca. 280 Ma that experienced a slab break-off event after ca. 260 Ma[4,8,27–30]. These works have also attributed variable roles to the gravitational collapse of the Early Permian fold and thrust belts along the southwestern Pangea margin[1,2,6,8,13–15,30], and the concomitant NE–SW tensional stresses linked to the prelude of the Pangea breakup[6,13,14,22]. Most recent studies have favored a subduction-related setting for part or the totality of the Choiyoi Magmatic Province based on the documentation of a geochemical magmatic arc signature that is locally diluted at the top of this SLIP showing characteristics of a within-plate or post-orogenic environment (e.g., [3–5,8,20,21,23,25,26,29,31,32]). In the intracratonic magmatic belt of this igneous province, this geochemical change is not documented, and a clear within-plate affinity is dominant[33,34] (Fig. 1b). The subduction-related hypothesis has been recently favored by δ$_{18}$O and Lu–Hf in zircons and Sr–Nd–Pb isotopic data indicating a shift from mainly crustal-derived sources in the pre-Choiyoi magmatism, to mantle-like and mixtures of these two end-members in the Choiyoi Magmatic Province[22,25,35–37].

An unappreciated record that may shed light on the origin of the Choiyoi Magmatic Province is the mantle structure. Improvements in tomographic techniques and the correlations of geological subduction records to imaged fossil slabs reveal that the lower mantle preserves information of ancient subduction configurations (e.g., [38–40]). As the tomographic visibility of the lower mantle lies around ca. 200–300 Ma[39,41–44], deep slab remnants provide an alternative and independent constraint to test the contrasting hypotheses for formation of this SLIP.

In this work, we evaluate these hypotheses by integrating the lower mantle structure corresponding to the southwestern Pangea margin as imaged by P-wave global seismic tomography[45] and Paleozoic plate-kinematic reconstructions using different reference frames[39,46–51]. Our reconstructions show a palaeogeographic coincidence between the Choiyoi Magmatic Province and a major slab gap in the lower mantle. This observation along with an analysis of a large geochemical dataset evidencing a slab break-

off magmatic signature in this SLIP indicates a large-scale mid-Permian-Early Triassic slab loss event. These results not only have implications for understanding the origin of one of the drivers of the most severe mass extinction on Earth, but also for the formation of SLIPs. Finally, this study illustrates an interdisciplinary approach to solve geological problems testing geodynamic hypotheses in ancient convergent settings back to the late Paleozoic.

## Results and discussion

**Linking deep slabs to the igneous record of the southwestern Pangea margin.** We analyze the lower mantle structure below the southwestern Pangea margin to a depth corresponding to the subduction configuration in the middle Permian to Late Triassic[42,43]. This was accomplished by overlapping the reconstructed positions of the southwestern Pangea margin at these times using the plate kinematic model of Matthews et al.[46], with tomographic slices from the UU-P07 P-wave global seismic tomography model[45] (see "Methods" section). The Paleozoic plate kinematic model of Matthews et al.[46] is based on Domeier and Torsvik[47]. The latter study implemented a paleomagnetic absolute reference frame corrected for true polar wander and built models with paleolongitude controls by considering the plume generation zone method[48]. Also, we carried out additional reconstructions of Pangea using the lower mantle slab reference frame of van der Meer et al.[39], the supercontinent orthoversion model of Mitchell et al.[49], and the Paleozoic plate kinematic model of Young et al.[50], which adopts a purely paleomagnetic reference frame[51] (Supplementary Fig. S1). The UU-P07 P-wave seismic model has been previously used to build plate reconstructions from late Paleozoic to Mesozoic and to identify deep fossil slabs worldwide[39,42,43]. As in previous tomotectonic studies[38–40,42,43,52], we assumed a vertical slab sinking and considered minor lateral migration in the lower mantle, by only ~100–200 km every 100 Myr[53]. Although, slabs can be dragged laterally if remain attached in plate collisions[54], this assumption has produced well-constrained tectonic models in several ancient convergent settings, where tomographically imaged slabs have been integrated consistently with geological data in plate margins[38–40,42,52,55,56]. A vertical slab sinking hypothesis is suitable for our analysis because the examined fossil slabs beneath the southwestern Pangea margin present wall-like geometries, which form in steeply dipping and quasi-stationary subduction zones[40] (Supplementary Fig. S2). Also, we considered a constant slab sinking rate in our reconstructions. Although slabs may experience a transient stagnation at the 660 km discontinuity and subduct at different velocities in the upper and lower mantle[43,44], as demonstrated in previous work[38–40,42,52,55,56] and our analysis, this assumption captures the overall subduction evolution and leads to a robust correlation between the mantle structure and the geological record. Furthermore, penetration of the mantle transition is facilitated by slowly retreating or stationary slabs[57] that form wall-like geometries[40] as those associated with the fossil slabs studied here (Supplementary Fig. S2).

We conducted an analysis of tomographic slices at depths of 2790, 2450, and 2250 km considering a whole-mantle slab sinking rate of 1 cm/yr, which is identical to optimal values determined in previous studies[38,40,42,52,55,56]. Hence, these depths would correspond to three representative subduction stages in the southwestern Pangea margin at ca. 280, 245, and 225 Ma (Fig. 2a–c). We carried out a broader analysis including further average slab sinking rates determined in additional studies (1.1 cm/year[58]; 1.2 cm/year[43]; and 1.3 cm/year[44]) (Supplementary Fig. S3). Noteworthy, the latter average values are within the range of lower mantle slab sinking rates considered the most

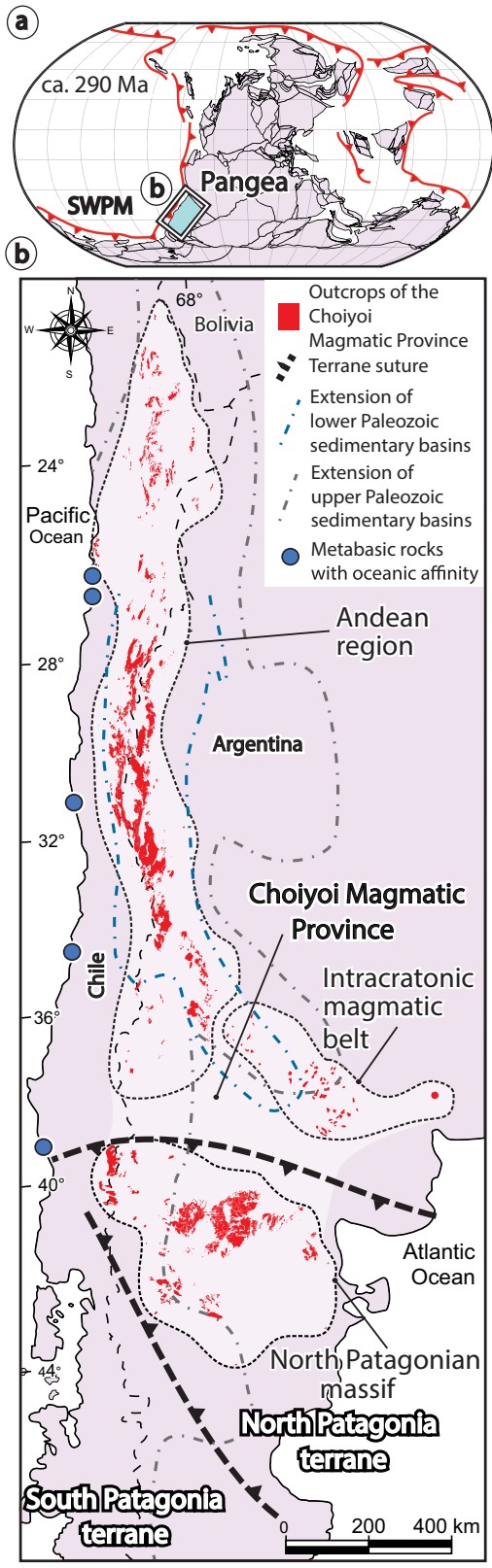

**Fig. 1 Location of the Choiyoi Magmatic Province within the Pangea supercontinent. a** Plate kinematic reconstruction of Pangea at 290 Ma from Matthews et al.[46]. **b** Schematic map in present day coordinates showing in red the distribution of the mid-Permian-Lower Triassic Choiyoi Magmatic Province after Rocher et al.[4]. Dashed black lines are the Late Paleozoic terrane sutures zones belonging to the North Patagonia[27] and the Southern Patagonia terranes[16]. Location of metabasic outcrops with oceanic affinities in the western continental margin is from Díaz-Alvarado et al.[81] and Hyppolito et al.[85] and references therein.

stacking different seismic tomography models at a specified mantle depth and detecting, where the models agree based on an increasing vote count. In high-velocity vote maps, higher vote counts help to highlight robust high-velocity structures across the analyzed models that are interpreted as the subducted oceanic lithosphere[58].

To better understand the origin of the Choiyoi Magmatic Province, we also compiled and analyzed a large geochemical dataset (total $n > 700$; filtered $n = 379$; see "Methods" section and supplementary Data file S1) including igneous rocks encompassing the three main geographical divisions of the Choiyoi Magmatic Province[3] corresponding to Andean region (ca. 286–247 Ma), the intracratonic magmatic belt (ca. 276–239 Ma), and the slightly older North Patagonian massif region (ca. 295–248 Ma) (Fig. 1b). The dataset also includes samples from the Late Carboniferous-Early Permian (ca. 330–290 Ma) and the Late Triassic-Jurassic (ca. 230–155 Ma) magmatic stages along the southwestern Pangea margin. The geochemical dataset is analyzed in recent tectonomagmatic discrimination diagrams that allow distinguishing slab break-off, arc, and within-plate magmatism[60,61] (see "Methods" section). To test previous hypotheses linking arc dynamics and extension to slab-rollback and slab steepening in the Andean region of the Choiyoi Magmatic Province[5,22,24,25,35], we also evaluate the late Paleozoic to Mesozoic spatiotemporal magmatic evolution. For this analysis, we used four previous compilations of available radiometric ages in the Andean region (U/Pb, Ar/Ar, K/Ar, Rb/Sr; $n = 122$) of Carboniferous to Lower Triassic igneous rocks from 21°S to 42°S from Navarrete et al.[62], del Rey et al.[5,22], and Ramos and Folguera[63] (supplementary Data file S2). In the spatiotemporal diagram, distance to igneous rock ages was plotted perpendicular to the current margin. These values were later corrected for subduction erosion considering a long-term (~70 Ma) rate of continental margin retreat equal to 1 km/Ma[64]. We did not consider shortening in the plate margin or possible strike-slip motion in the forearc region, and hence, plotted arc-to trench distances represent a minimum value.

The tomotectonic analysis shows two main high-velocity anomalies beneath the reconstructed southwestern Pangea margin previously interpreted by van der Meer et al.[39,43] as fossil slabs reaching the core-mantle boundary (Fig. 2a–c). Both high-velocity anomalies are located between ~1950 and 2350 km from the current Andean trench (Fig. 2d). The anomaly beneath the northern area of the southwestern Pangea margin is referred to as the São Francisco anomaly and has been associated with ca. 245–225 Ma arc magmatism that took place in an Andean-type margin[43]. Recent geological studies by Spikings et al.[65] and Chew et al.[66] indicate a longer subduction record in this area linked to arc magmatism and metamorphism between ca. 290 and 220 Ma, attesting to protracted subduction associated with the formation of the São Francisco slab wall. The other mantle structure is located in the southern area of the reconstructed southwestern Pangea margin below Patagonia, South Africa, and the Antarctic Peninsula, and is referred to as the Georgia Islands anomaly[43].

compatible with the surface geological record linked to the Meso-Cenozoic subduction history beneath South America (1 cm/year[52]; ~1–1.1 cm/year[56]; and ~1.3 cm/year[59]) (Supplementary Fig. S2). To identify consistent mantle structures resolved in several seismic tomography models, we built lower mantle vote maps that allow a straightforward comparison of multiple models[58] (see "Methods" section). Vote maps are generated by

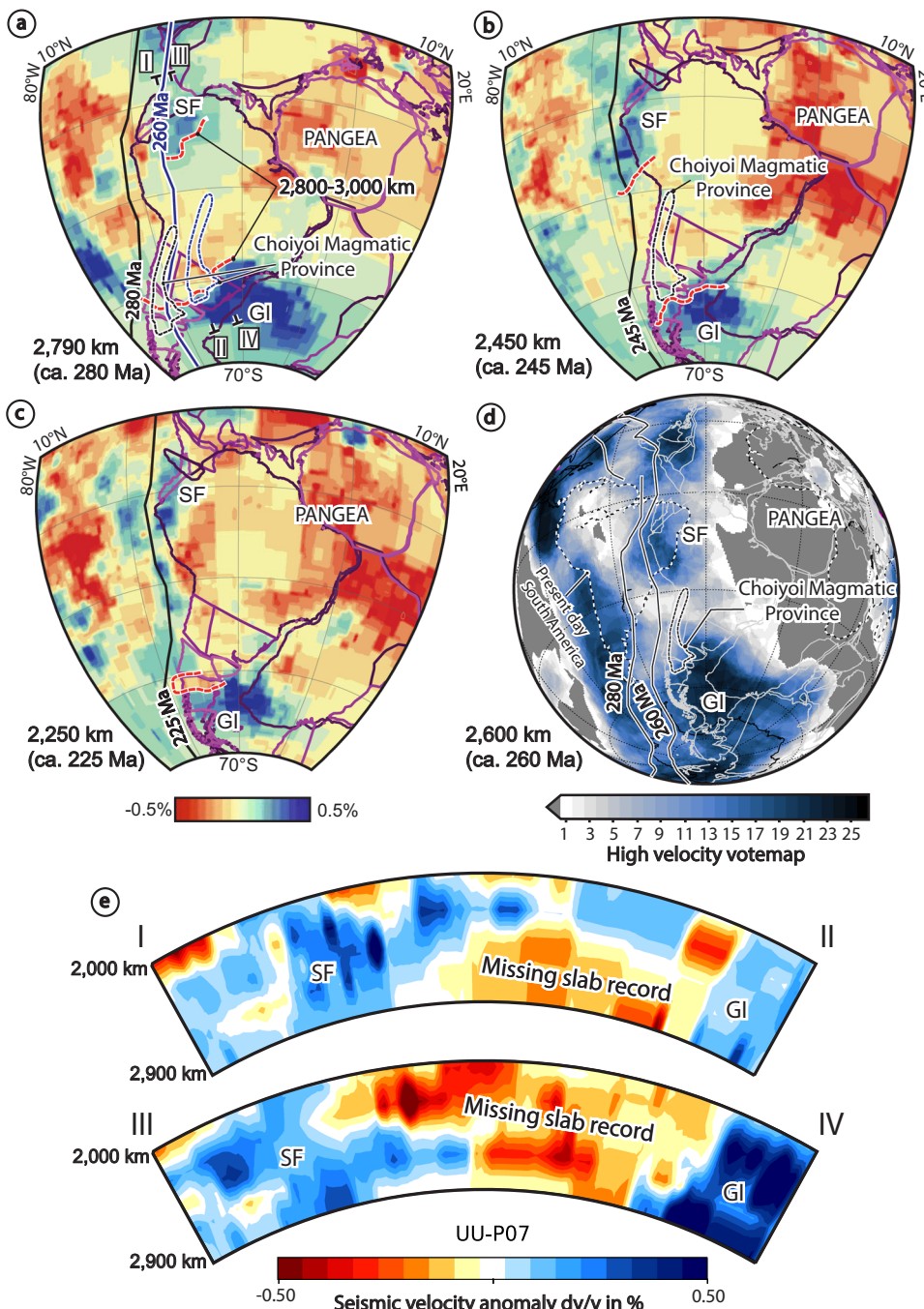

**Fig. 2 Tomotectonic analysis of the southwestern Pangea margin.** Mantle tomography slices of the UU-P07 seismic tomography model[45], showing the Georgia Islands and São Francisco slabs[43] at **a** ca. 280 Ma, **b** ca. 245 Ma, and **c** ca. 225 Ma considering a lower mantle slab sinking rate of 1 cm/year[38,40,42,52,55,56]. Overlaid are plate reconstructions[46] at **a** 280 and 260 Ma, **b** 245 Ma, and **c** 225 Ma. Dotted red line indicates a high-velocity discontinuity interpreted as a major slab gap along the southwestern Pangea margin. **d** High-velocity vote map stacking 26 global seismic tomography models at 2600 km with overlapped plate reconstructions[46] showing the consistency of the lower mantle structure and the high-velocity discontinuity along-strike the reconstructed southwestern Pangea margin. Thick solid lines labeled with ages west of South America in **a**–**d** represent reconstructed trench positions after the restoration of Cenozoic Andean shortening[46]. GI Georgia Islands slab, SF São Francisco slab. **e** Cross-sections of the UU-P07 tomography model indicating the area with a missing slab record. See profiles location in **a**.

This anomaly has been interpreted as a fossil slab wall associated with arc and metamorphic records spanning from ca. 290–185 Ma in the western Patagonia-Antarctic Peninsula segment of the southwestern Pangea margin ([43,52,62] and references therein) (Fig. 2).

Reconstructions in Fig. 2a, b with tomographic mantle depths corresponding to the Permian-Early Triassic subduction stage

indicate a first-order segmentation of the southwestern Pangea margin. A vote map including the analysis of 26 global P-wave and S-wave tomography models at a representative depth of 2,600 km confirms the high-velocity discontinuity along the southwestern Pangea margin (Fig. 2d). Information about the seismic tomography models used for this analysis is presented in supplementary Table S1 and additional vote maps at different

lower mantle depths are presented in supplementary Figure S4. Cross-sections indicate a lateral extent for this discontinuity of ~2,000 km (Fig. 2e). The reconstructed size along these cross-sections, achieved by a surface projection of the lower mantle velocity discontinuity in our tomotectonic reconstructions, indicates an originally larger size of ~2,800-3,000-km (Fig. 2a). Notably, the high-velocity discontinuity coincides with the reconstructed position of the Choiyoi Magmatic Province (Fig. 2a,b). We consider that the lateral extent of this high-velocity discontinuity is not compatible with a local thermal anomaly within the slab and hence, it is here interpreted as a large-scale slab gap. The slab discontinuity mostly disappears at mantle depths corresponding to the ca. 230-225 Ma subduction stage (Fig. 2c). Additional tomotectonic analyses implementing plate kinematic reconstructions with different reference frames[39,49,51] yield similar results indicating a robust connection between the mantle structure and the southwestern Pangea margin in the late Paleozoic (Supplementary Fig. S1). The only exception is the Triassic stage in reconstructions implementing the orthoversion model of Mitchell et al.[49], where the southwestern Pangea margin is reconstructed to the east of the lower mantle slabs. Although not highlighted nor explained in terms of geodynamics, this high-velocity discontinuity can also be observed in previous global tomotectonic analyses[39].

The timing and origin of the slab gap can be further constrained by the ages and geochemistry of the Choiyoi Magmatic Province. Tectonic discrimination diagrams designed to distinguish between magmatic arc and slab break-off geodynamic contexts and tested in ancient and current convergent settings[60,61] indicate a clear affinity with slab break-off magmatism in the three regions of the Choiyoi Magmatic Province (Fig. 3 and Supplementary Figs. S5 and S6). The La/Yb$_N$ vs. Yb$_N$ diagram[67] shows that part of the magmatism classifies as adakitic, which is particularly evident for the North Patagonian Massif region and the intracratonic magmatic belt (Supplementary Fig. S7). The preceding Late Carboniferous-Early Permian magmatism is distributed in the arc and slab break-off fields. However, a dominant tendency towards the magmatic arc field is observed in this case (Supplementary Fig. S8). Also, the Upper Triassic-Jurassic igneous record exhibits a strong tendency for the magmatic arc field (Supplementary Fig. S9). Thus, the geochemistry of the Choiyoi Magmatic Province is compatible with the development of a slab gap likely resulting from slab break-off events between ca. 286 and 247 Ma as indicated by the lifespan of this SLIP. However, this process seems to have begun locally earlier at ca. 295 Ma in the North Patagonian Massif (Fig. 3 and Supplementary Figs. S5 and S6). The pre-Choiyoi and post-Choiyoi Magmatic Province magmatism are compatible with subduction processes (Supplementary Figs. S8 and S9). The spatiotemporal analysis of the Late Carboniferous-Early Triassic magmatic activity in the Andean region reveals an eastwards magmatic expansion of the Choiyoi Magmatic Province between ca. 286 and 247 Ma (Fig. 4). During this stage, magmatic activity in the Andean region reached as far as ~560 km from the reconstructed paleo-trench axis and broadened up to ~300 km.

Additional tomotectonic analyses including a slab sinking rate of ~1.1 cm/year[58], within the range of previous estimates obtained from a similar analysis of subduction zones[40,68], yield a scenario compatible with the slab break-off magmatism of the Choiyoi Magmatic Province, but with a preserved slab gap up to ca. 265 Ma (Supplementary Fig. S3). Reconstructions with faster lower mantle slab sinking rates shift this evolution towards younger times and are not compatible with the clear arc geochemistry in the igneous record along the southwestern Pangea margin in Late Triassic-Jurassic times indicated in previous studies[25,31,69] and confirmed in our geochemical analysis (Supplementary Fig. S9). Independent of the assumed slab sinking rates, the tomotectonic analysis reveals the presence of a large-scale slab-free area beneath the reconstructed southwestern Pangea margin.

**The origin of the Choiyoi Magmatic Province.** To test the competing hypotheses for the origin of the Choiyoi Magmatic Province, we have linked the surface and mantle slab records through plate-kinematic reconstructions and analyzed geochemical data and the spatiotemporal igneous record. Our tomotectonic maps reveal a ~2800–3000 km-wide high-velocity discontinuity along the southwestern Pangea margin, documenting a palaeogeographic coincidence between the mid-Permian-Lower Triassic Choiyoi Magmatic Province and a slab-free area (Fig. 2). This spatiotemporal co-occurrence is also observed in reconstructions that employ alternative reference frames with different paleo-longitude constraints[39,49] and average lower mantle slab sinking rates[42,44,58] that are compatible with the younger subduction evolution beneath the South American-Caribbean region[52,55,56] (Supplementary Figs. S1, S2, and S3). This observation in conjunction with a marked slab break-off geochemical signature in the Choiyoi Magmatic Province indicates that SLIP emplacement was caused by a large-scale slab loss event (Fig. 3). Slab thickening during transit in the lower mantle, expected to have taken place in the São Francisco and Georgia Islands slab walls[43], could have narrowed the imaged lower mantle slab gap. However, thermal erosion at slab edges possibly partially compensated for this effect[70]. We suggest the recent proposal to exclude the intracratonic magmatic belt from the Choiyoi Magmatic Province based on a within-plate geochemical signature[34], must be reconsidered. Our results indicate a shared slab break-off geochemical signature in the Andean region, the intracratonic magmatic belt, and the North Patagonian massif, suggesting a common origin for the three geographic areas of this SLIP (Fig. 3 and Supplementary Figs. S5 and S6). In general, these findings are not compatible with a subduction-related setting for the totality or part of the Choiyoi Magmatic Province[3–5,8,20–26,29–31]. The recognition of a Permian-Triassic slab beneath southern South America and the Antarctic Peninsula[39,43] reinforces a subduction-related origin as suggested for the Permian magmatism in that area[18,37] (Fig. 2 and Supplementary Fig. S1). The latter does not support extension of the Choiyoi Magmatic Province into Antarctica[11], where age equivalent but significantly less voluminous calc-alkaline igneous rocks have been previously described[71].

For the Andean region, which comprises the most extensive area of the Choiyoi Magmatic Province, previous studies suggested a slab rollback and/or steepening scenario and subsequent mantle upwelling to explain the prevailing extensional regime and the increasing mantle imprint on magmatism[5,22,24,25,35]. However, in addition to our results which challenge major subduction, the record of an up to ~560 km inland magmatic migration is not compatible with the typical syn-extensional trenchward arc retreat produced by slab rollback and steepening (e.g., [62,63]) (Fig. 4). If the intracratonic magmatic belt is considered part of the Choiyoi Magmatic Province, as indicated here and in previous studies[2,33], the latter magmatic migration would have reached up to ~1000 km from the paleo-trench (Fig. 1b). Alternatively, we interpret this magmatic expansion as the result of the opening and eastward enlargement of the slab gap associated with the Panthalassa slab destruction.

The slab break-off geochemical signature is interpreted as produced by the partial melting of eclogite in subducted oceanic crust at depths of >2 GPa during slab detachment[60]. As melts ascend through the upper-plate they may be contaminated and if the mantle lithosphere is enriched they may develop crustal-like Sr and Nd isotopic signatures[60,61]. An alternative explanation is

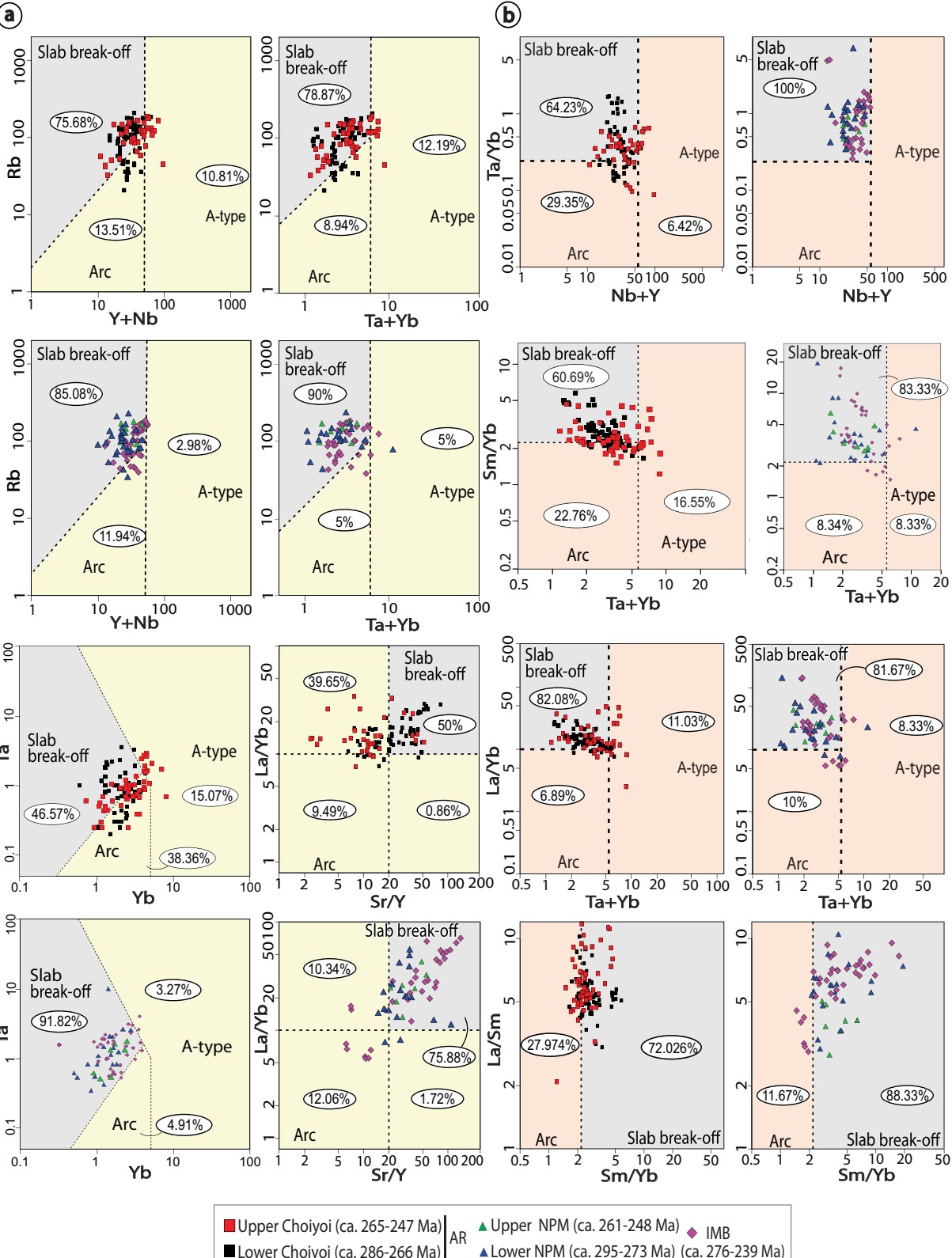

**Fig. 3 Geochemical diagrams to discriminate magmatic arc, slab break-off, and within-plate tectono-magmatic settings in middle Permian-Lower Triassic igneous rocks of the Choiyoi Magmatic Province. a** Hildebrand et al.[60] and **b** Whalen and Hildebrand[61]. Geochemical data is available in Data file S1 in supplementary information. AR Andean region, IMB Intracratonic magmatic belt, NPM North Patagonian massif.

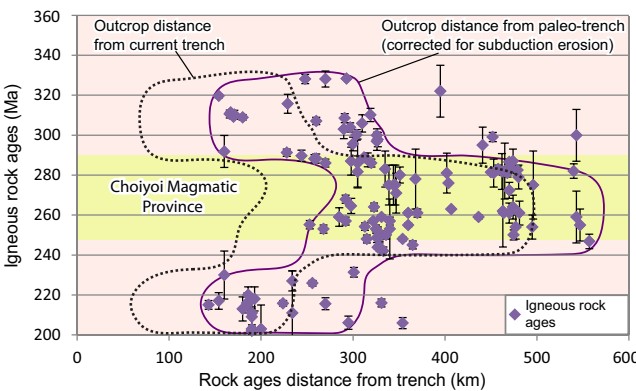

**Fig. 4 Spatiotemporal evolution of the Late Paleozoic-Early Mesozoic magmatic activity in the Andean region segment of the Choiyoi Magmatic Province (21°S–42°S)**[5,22,62,63]. This analysis reveals an eastwards magmatic migration and broadening of the Choiyoi Magmatic Province between ca. 286 and 247 Ma. Geochronological data is available in supplementary Data file S2.

the differentiation of high-K mafic magmas produced by melting of an enriched lithospheric mantle within the stability field of garnet with heat supplied by slab break-off-related and extension-related asthenosphere upwelling[72]. These models are not mutually exclusive, in either scenario the rocks generated by slab break-off are enriched in Sr, Nb, Ta, and Eu, and depleted in the HREEs and Y relative to arc rocks and exhibit variable isotopic signatures[60,61] (see "Methods" section). The fact that most of the Choiyoi Magmatic Province was emplaced during an extensional-transtensional tectonic regime that collapsed the Lower Permian orogenic crust[4–8,25,33,34] precludes attributing rocks with high La/Yb, Sr/Y, Gd/Yb, and Sm/Yb ratios to partial melting of a thickened lower crust or high-pressure fractional crystallization (e.g., [73]). This is particularly evident for the intracratonic magmatic belt, where the above-mentioned geochemical characteristics appear in rocks emplaced in a region with normal crust thickness and negligible Permian shortening[33,34]. In orogenic settings resulting in significant crustal thickness, such as in the Central Andes, high and low values of the ratios discussed above are equally common, as fractional crystallization and crustal assimilation can also occur at shallow crustal depths[74].

Underplating of these melts would have also prompted melting of a hydrous mafic lower crust inherited from previous subduction stages[3,22,75] and magma hybridization[5,20,29,34,35]. Incorporation of older continental crust and arc basement is evident by the presence of inherited zircons (e.g., Andean region 420–1440 Ma[35]; North Patagonian massif region 320–1000 Ma[16]). Increased extensional activity in the Andean region during the upper section of the Choiyoi Magmatic Province (ca. 265–247 Ma)[4–7,21], favored asthenospheric decompression melting increasing the volume of the underplated melts and the mantle input[5,22,25,35–37]. As previously suggested for this SLIP, upper mantle warming produced by supercontinental thermal insulation may have acted as an additional heat source enhancing partial melting in this region[1]. According to the geochemical analyses of MORB basalts formed immediately after the Pangea break-up, this process resulted in upper mantle temperatures 150 °C higher than the present-day average[76].

The origin of the large-scale slab loss event in the southwestern Pangea margin is intriguing. In the North Patagonian Massif segment of the Choiyoi Magmatic Province south of 39°S, Permian-Triassic slab break-off events have been proposed after the Late Carboniferous accretion of the Southern Patagonia terrane[16] and the Early-Middle Permian accretion of the North Patagonia terrane to the southwestern Pangea margin[27] (Figs. 1b

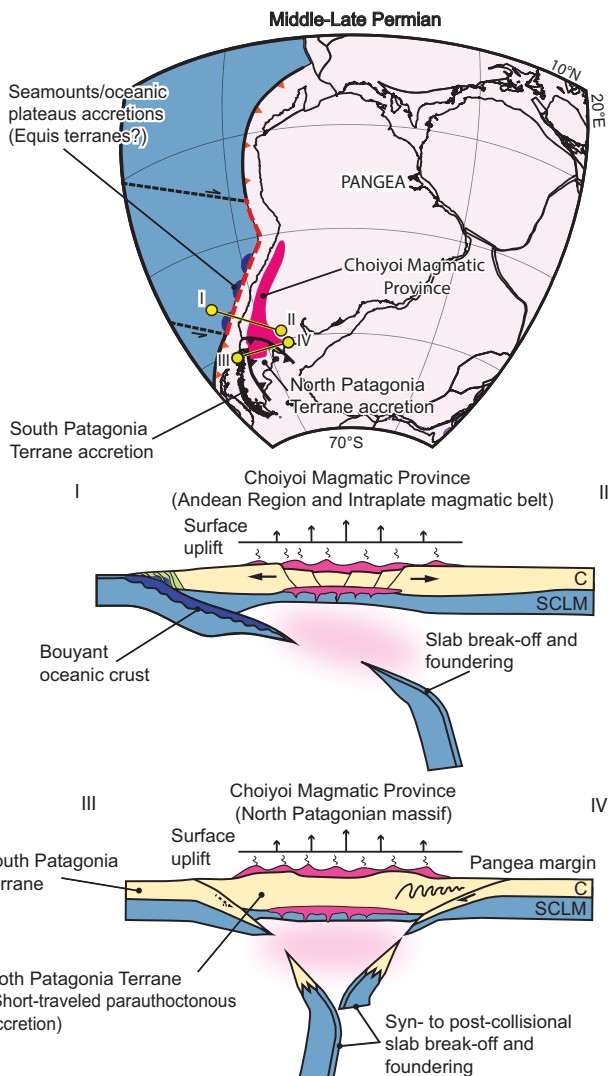

**Fig. 5 Conceptual model for the development of the Choiyoi Magmatic Province in the southwestern Pangea margin.** Schematic plate kinematic reconstruction in middle-Late Permian times illustrates the geodynamic processes that triggered slab break-off events and the development of the Choiyoi Magmatic Province in southwest Pangea. Profile I–II shows the geodynamic model for the Andean region and the intracratonic magmatic belt of the Choiyoi Magmatic Province with slab break-off triggered by buoyant oceanic crust accretion at different latitudes. Profile III–IV illustrates the geodynamic model for the North Patagonian massif region of the Choiyoi Magmatic Province where slab break-off results from continental terrane accretions. Abbreviations in profiles are; C Continental crust and SCLM Sub-continental lithospheric mantle.

and 5). The Southern Patagonia terrane formed a continental microplate separated from Gondwana in the Cambrian that was accreted back to the margin in the Mid-Carboniferous[16]. The origin of the North Patagonia terrane is a topic of a longstanding debate for which several hypotheses have been put forward. The allochthonous hypothesis suggests that this continental terrane was attached to East Antarctica and drifted away in the Silurian[27]. Following subduction interpreted as recorded by Permian magmatism of the North Patagonian Massif, the continental terrane was accreted to the southwestern Pangea margin in the Early-middle Permian[9,27]. Contrasting studies have suggested that the North Patagonia terrane is autochthonous to Gondwana and that the subduction-related Permian magmatism instead

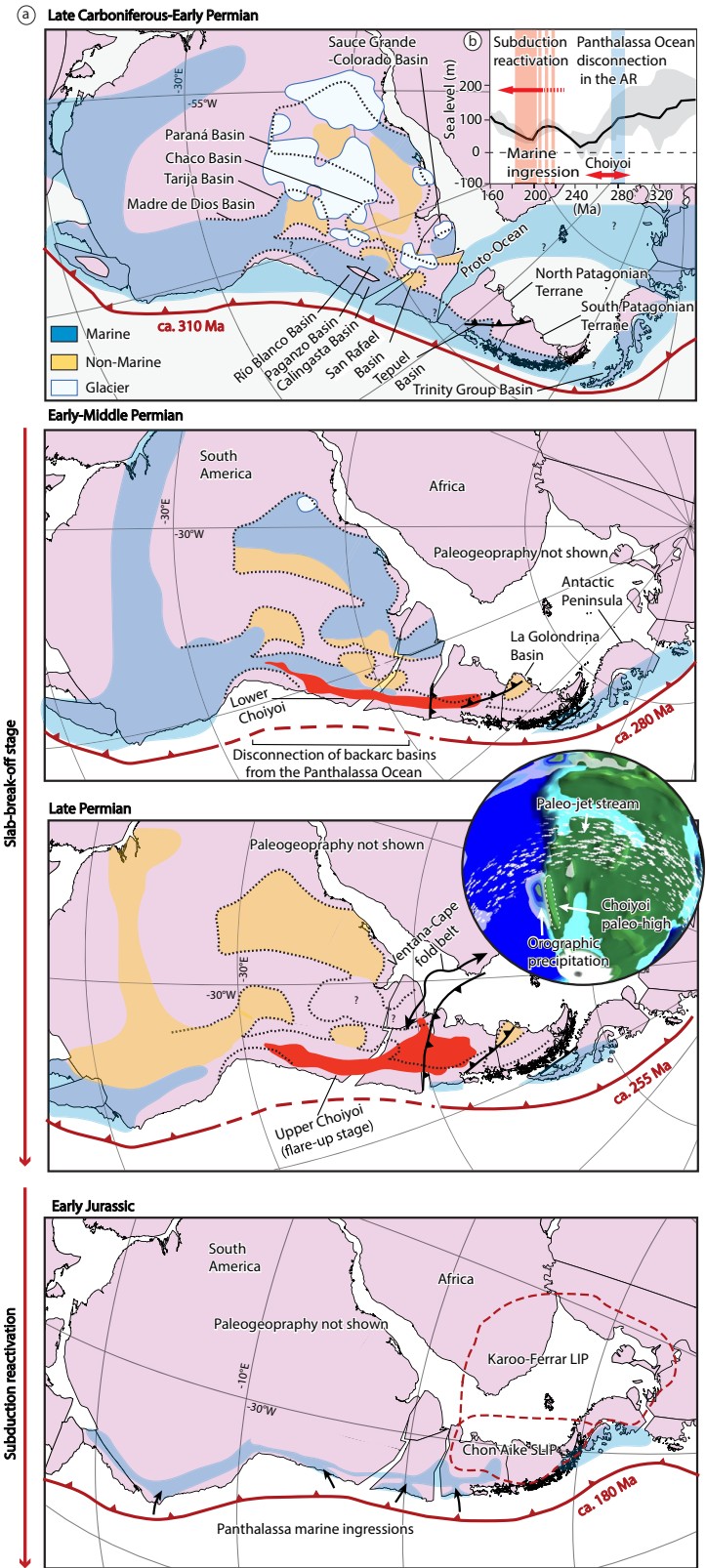

represents slab break-off associated with the previous accretion of the southern Patagonia terrane to the southwest[16,18]. The parautochthonous hypothesis suggests that the North Patagonia terrane was separated by rifting during the opening of an early Paleozoic oceanic basin that was closed in the middle-Late Permian. This is based on paleomagnetic evidence indicating a

potential separation ≤1500 km of this terrane in pre-Permian times[77]. Recently, a short-traveled parautochthonous hypothesis has been favored based on new paleomagnetic poles indicating a reconstructed position of the North Patagonia terrane from ca. 450–250 Ma close to its current location[78]. In this model, the North Patagonia terrane was located close to the Pangea margin

**Fig. 6 Late Carboniferous to Early Jurassic paleogeographic reconstructions of southwestern Pangea. a** Maps illustrating the paleogeographic modifications of the plate margin before, during, and after the emplacement of the Choiyoi Magmatic Province. These maps show the spatiotemporal relation between the emplacement of the early magmatic activity of the Choiyoi Magmatic Province and the marine disconnection between the Panthalassa Ocean and the backarc basins in Pangea and the Late Permian continentalization during the Choiyoi magmatic flare-up[3,92,93]. These processes are here interpreted as triggered by surface uplift ≥1 km caused by the large-scale slab-break-off associated with the emplacement of the Choiyoi SLIP. The maps also illustrate the close spatiotemporal relation between the subduction restoration in the Late Triassic-Jurassic and the recovery of marine ingressions in the Pangea margin[98]. The latter is here interpreted as produced in part by the reactivation subduction-related dynamic subsidence in this area. Inset in Permian maps is a paleoclimatic simulation at the Kungurian (279.3–272 Myr)[95] reproducing orographic precipitation associated with the Choiyoi paleotopography. **b** Averaged eustatic sea-level curve from Carboniferous to Jurassic times[96] with references to the geological events discussed in this study. Gray envelope corresponds to the extrema.

in the westernmost area and was separated to the east by a 200–600 km-wide V-shaped basin[78] (Fig. 6a). Mid-Permian silicic magmatism in the region coexisted in part with shortening associated with the accretion of one or both terranes[16,27] (Fig. 1b). Syn-magmatic transpression in the lower section of the Choiyoi Magmatic Province was locally described in the southernmost Andean region[8], likely indicating the far-field influence of these deformational events. Our results have direct implications regarding the origin of the North Patagonia terrane. The lack of magmatic arc affinity in the Permian-Triassic igneous rocks of the North Patagonian Massif and the overall slab break-off magmatic signature in the area are not compatible with an allochthonous North Patagonia terrane[27] (Fig. 3 and Supplementary Figs. S5, and S6). These observations also negate a far-traveled para-autochthonous origin for this terrane (i.e., separation ≤ 1500 km previous to accretion[77]), which necessitates a pre-accretion subduction stage as well. One possibility is that the North Patagonia terrane was autochthonous in the Permian and slab break-off magmatism in the North Patagonian Massif resulted from slab detachment after the Late Carboniferous accretion of the Southern Patagonia terrane[16,18] (Fig. 1b). Alternatively, the North Patagonia terrane could have been a short-traveled para-autochthonous domain[78], where the lack of arc magmatism before the terrane accretion and the subsequent slab break-off magmatism would suggest the closure of a narrow proto-oceanic basin. In this context, a short proto-oceanic lithosphere, which upon subduction would have not reached the critical slab dehydration depth (120 ± 40 km[79]), would explain the lack of arc magmatism before the accretion process. A short slab would have not been dehydrated enough to trigger substantial hydrous melting of a mantle wedge, thus precluding magmatic arc development. Nevertheless, it would have still provided enough negative buoyancy to eventually detach, driving syn-accretion to post-accretion slab break-off magmatism[60,61]. As proposed for the Pyrenees and Alpine pre-collision stage, the lack of subduction-related magmatism in the North Patagonian massif could also be explained by the subduction of a hyperextended margin and a dry oceanic mantle lithosphere[80]. In this way, both accretionary processes would have been involved in the slab loss event in the North Patagonian Massif region of the Choiyoi Magmatic Province (Fig. 5).

For the Andean region of the Choiyoi Magmatic Province, Mpodozis and Kay[15] suggested a slab break-off event associated with the Early Permian accretion of the Equis terrane. Previous studies have argued against this hypothesis based on the lack of concrete evidence of an allochthonous basement in this area[22,25,81]. Although speculative, evidence of the Equis terrane could be absent due to subsequent forearc subduction erosion in the Andes[15,64]. According to Pankhurst et al.[16], an alternative to the Equis terrane hypothesis is a fast northward propagation of a rupture in the slab initially induced by the Southern Patagonia terrane collision. Alternatively, this process could have taken place in an ocean-continent subduction setting, as suggested

locally throughout the Andean region[8,14,17]. A potential explanation is slab detachment caused by the subduction of a mid-ocean ridge leading to the formation of a slab window[28,70]. The emplacement of alkaline to tholeiitic backarc plateaus or calc-alkaline rocks with intraplate geochemical signatures would be diagnostic of slab window development (e.g., [70,82]). This typical slab window record in the backarc area is not observed; instead, the Choiyoi Magmatic Province comprises large volumes of mesosilicic to silicic magmatism with slab break-off geochemical signature (Fig. 3). In subduction settings, the interaction of a buoyant oceanic feature (e.g., oceanic plateau or seamount) can drive slab break-off of the oceanic lithosphere and the formation of large slab gaps[83], with subduction reestablishment demanding variable times and up to ~40 Ma depending on plate-kinematics[84]. A diachronous accretion of oceanic topographic features between ~300 and ~270 Ma from 26°S to 39°S is documented in relicts of accretionary prisms along the Andean margin[17,81,85,86] (Fig. 1b). These mafic rocks share similar geochemical and isotopic characteristics from northern to south-central Chile, involving an enriched mantle source linked to plume activity or plume-ridge interactions[81,85] (Fig. 2). Hence, one possibility is that the accretion of these thickened/buoyant oceanic features would have initiated local slab gaps[83] that later coalesced and propagated laterally, driving a massive slab loss in the Andean region of the Choiyoi Magmatic Province (Fig. 5). The incipient formation of slab gaps could have begun in Late Carboniferous-Early Permian times, as indicated by the early arrival of oceanic features at around 300 Ma to the west of the Andean region[81,85] and the detection of some influence of slab break-off magmatism at this time (Supplementary Fig. S8). We suggest that a deeper slab detachment in a context of shallow subduction between 35°S and 40°S would explain the easternmost location of the slab break-off magmatism in the intracratonic magmatic belt and the local persistence of an accretionary prism[8,87,88] (Fig. 1b). This context supports the notion of a series of Equis terranes formed by several oceanic pieces accreted to the southwestern margin of Pangea instead of the docking of a single continental fragment. As suggested by numerical models and tomographic observations[89,90], after slab break-off the detached slab fragments likely sunk faster in the mantle than the São Francisco and Georgia Islands slabs. As small fragments are more effectively heated by conduction, at the core-mantle boundary, these relict slabs would have been assimilated or become thermally invisible. Subduction activation along most of the southwestern Pangea margin may have taken place since ~230–220 Ma. This is indicated by the progressive recovery of the slab gap in the tomotectonic analysis and the magmatic arc signatures in the Upper Triassic-Jurassic rocks from the Andean region of the Choiyoi Magmatic Province (Fig. 2c and Supplementary Fig. S9).

Therefore, the Choiyoi Magmatic Province would have resulted from continental terrane collisions during the assembly of Patagonia[16,27] and the accretion of buoyant oceanic highs to

north of 40°S[17,81,85,86]. These processes would have jointly triggered a large-scale destruction of the subducting slabs (Fig. 5). Except for a few older magmatic ages between ~295 and 290 Ma in the North Patagonian Massif[16], the predominant age range between ~286 and 247 Ma[2,3] indicates roughly simultaneous slab break-off processes in the three geographic areas of this SLIP. The reduction in plate margin tectonic stresses caused by the widespread slab loss event, the upper mantle warming produced by supercontinent thermal insulation, and the first-order global tensional stresses associated with the incipient breakup of Pangea[1,6,13,20] may have jointly promoted extension, orogenic collapse, and protracted magmatism.

**Implications for Permian climate change, paleogeography, pre-Cenozoic SLIPs, and slab break-off processes.** Geodynamic changes during the mid-Permian-Early Triassic recorded in the igneous record and the fossil mantle structure had an impact beyond the tectono-magmatic evolution of the southwestern Pangea margin. Previous studies suggested that extensive magmatic intrusion in organic-rich shales, peat, and carbonates that accumulated in early Paleozoic backarc basins triggered the release of large volumes of $CO_2$ and $CH_4$ into the Permian atmosphere[5,10] (Fig. 1b). The coeval emplacement of LIPs (e.g., Emeishan and Siberian Traps LIPs) amplified global warming driving Earth's most severe extinction[5,10–12]. Although the connection between the Choiyoi Magmatic Province and Permian global warming and extinction has been well-established, the role of magmatic migration and broadening in triggering this process has been overlooked. Contrary to slab shallowing that produces inland arc migration with a decreasing magmatic volume[63], slab break-off processes drive inland migration of voluminous magmatic activity[91]. We suggest that the ~560–900 km magmatic migration triggered by slab loss in the Andean and intracratonic magmatic belt regions was a key process allowing distal magma intrusion into organic-rich Paleozoic basins inland southwest Pangea (Figs. 1b and 4). Otherwise, magmatic intrusion during typical subduction-related stages[25,79] would have been less extensive and progressive, and hence, the impact on climate change, probably weaker.

The Choiyoi Magmatic Province also had a major impact on the paleogeographic evolution of southwestern Pangea. The emplacement of this SLIP produced profound modifications in the late Paleozoic plate margin along the Andean region in partially or completely closing the marine connection of the backarc basins with the Panthalassa Ocean[92] (Fig. 6a). The magmatic activity was accompanied by the progressive development of an extensive orographic barrier in the Late Permian that divided the western margin of Pangea into a western temperate region under marine influence and an inland eastern region, where semiarid and arid conditions prevailed[93] (Fig. 6a). The positive relief would have forced the rise of humid winds from the Panthalassa Ocean, trapping their moisture in the coastal region and enhancing aridity in the continental interior[92,93]. The Choiyoi orographic barrier must have been ≥1 km, which is the minimum height to develop a noticeable rain shadow effect (e.g., [94]). The orographic precipitation induced by the Choiyoi topography is reproduced in recent paleoclimatic simulations including paleo-DEM reconstructions[95] (see "Methods" section) (Fig. 6a).

How this positive topography was formed along the Andean region is still poorly understood. Furthermore, it is difficult to explain why backarc basins were disconnected from the Panthalassa Ocean during a period of relatively high global sea level[96] (Fig. 6b). Our findings address these challenges. Conceptual and numerical studies indicate that slab break-off can drive rapid surface uplift above ~1 km[19,97]. Numerical models exploring the topographic response of slab break-off indicate surface uplift at rates between 0.1 and 0.8 km/Ma with post-break-off uplift lasting between ~10 and 20 Myr[97]. The resultant topographic uplift from massive slab break-off linked to the Choiyoi SLIP provides an explanation for both the disconnection of backarc basins from the Panthalassa Ocean and the Permian orographic barrier (Fig. 6a). We note that the subduction reactivation stage in the Late Triassic-Jurassic coincides with renewed marine inundation in the southwestern Pangea margin (Fig. 6a). The marine ingression began in the Late Triassic and is widely recognized since the Early Jurassic[98]. This is often explained by increased plate margin subsidence produced by backarc extension or thermal subsidence in a context of relatively high global sea level[14,98] (Fig. 6b). Plate margin inundation could have been enhanced by the reactivation of subduction-related dynamic subsidence. This process commonly produces about 1 km of downwarping in the upper-plate that facilitates marine ingression[99]. The influence of dynamic subsidence at this time is inferred from the close spatiotemporal relation between subduction reactivation and marine ingression (Figs. 2c and 6a, b).

Our results provide new avenues to assess the origin of SLIPs dating back to the Permian. SLIPs are less abundant in the geological record than mafic large igneous provinces and hence, indicate exceptional conditions to drive large volumes of silicic magmatism[100]. SLIPs are often restricted to continental margins that contain a fertile, hydrous lower crust built up by preceding subduction stages[1,3,100]. Previous studies have agreed on the requirement of upper-plate extension to drive SLIPs, but the precise geodynamic context is still widely debated[101,102]. SLIPs have been associated with intraplate extension linked to continental breakup (e.g., the Lower Cretaceous Whitsunday SLIP[100]), to active backarc convergent settings (e.g., the Cenozoic Sierra Madre Occidental SLIP[103] and Taupo volcanic zone[100]), and to arc settings (e.g., Upper Cretaceous Okhotsk-Chukotka Volcanic Belt[102]). Similarly to the ongoing debate about the geodynamic setting of the Choiyoi Magmatic Province, the origin of other SLIPs remains under debate (e.g., Jurassic Chon Aike SLIP: Plume-related model[104] vs. subduction-related model[105]). Difficulties arise when assessing the geodynamic setting of SLIPs in part due to the calc-alkaline nature of silicic rocks. Discrimination between original subduction-related signatures from those inherited from partial melting of pre-existing arc basement is notoriously challenging[100]. We suggest that the methodology followed in this study provides new means to test the end-member models for post-Carboniferous SLIPs. The presence/absence of ancient slabs beneath reconstructed margins can provide independent evidence to constrain active/inactive subduction during SLIP development. To illustrate this, we carried out a simple tomotectonic analysis for the three pre-Cenozoic SLIP examples mentioned above (Fig. 7 and Supplementary Fig. S10). We followed the same procedure and assumptions considered in our preferred reconstructions for the Choiyoi Magmatic Province. For the Jurassic Chon Aike and the Upper Cretaceous Okhotsk-Chukotka SLIPs, the reconstructed margins are located nearby or above high-velocity anomalies (Fig. 7a, b and Supplementary Fig. S10). We interpret this as evidence of ongoing subduction during the development of both SLIPs, supporting studies attributing variable roles to subduction for their origin[62,102,105]. For the Whitsunday SLIP, the eastern Australian margin is located in an area with no evidence of major high-velocity anomalies that could be interpreted as slab relicts (Fig. 7c and Supplementary Fig. S10). This is consistent with an origin linked to plate-margin extension associated with continental break-up[100]. Nevertheless, the absence of high-velocity anomalies in this region could also result from slabs lying at mid-mantle depths, where the resolution of tomography models in the southern hemisphere is degraded relative to other regions[58].

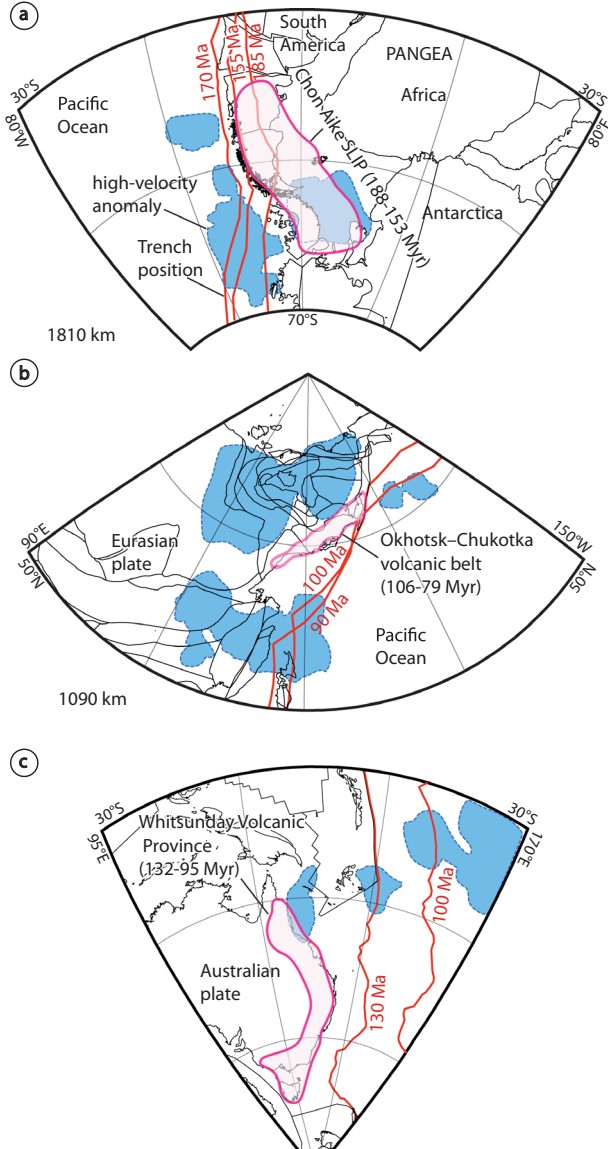

**Fig. 7 Interpretation of tomotectonic analysis of three different pre-Cenozoic SLIPs. a** Jurassic Chon Aike Magmatic Province[104], **b** Upper Cretaceous Okhotsk-Chukotka volcanic belt[102], and **c** Lower to Upper Cretaceous Whitsunday volcanic province[100]. The mantle structure in **a** and **b** support a context with active subduction during Chon Aike and Okhotsk-Chukotka SLIPs, whereas in **c** an arrested subduction setting is inferred for the Whitsunday SLIP. Mantle tomography slices are from the UU-P07 seismic tomography model[45] and overlaid plate reconstructions are from Mathews et al.[46]. The original tomotectonic analysis is found in Supplementary Fig. S10.

The findings presented in this study have implications for the role of slab break-off in the development of SLIPs associated with convergent settings. Although localized slab break-off has been proposed in the last stages of the Sierra Madre Occidental Province[103], SLIP development produced entirely by massive slab break-off, as documented in the Choiyoi Magmatic Province, is unprecedented. The latter would explain differences in lithological composition, geochemical signature, and the anomalous amount of intermediate (andesitic) rocks in the lower portion of the Choiyoi Magmatic Province contrasting with typical SLIPs[3], where rhyolitic magmatism is more abundant and exhibits within-plate or arc-like signatures[100,103]. We suggest

that slab break-off SLIPs are rather exceptional, possible only when multiple slab break-off triggers act in concert, as for the Choiyoi Magmatic Province (Fig. 5). These events would also be favored during supercontinent assembly or breakup stages when thermal insulation elevates upper mantle temperatures and enhances partial melting[1,76]. If we accept that plate tectonics was an ongoing process in the Archean-Paleoproterozoic, voluminous slab break-off magmatism would have been facilitated by weaker slabs, thicker oceanic crust, and mantle potential temperatures in excess of 150–250 °C at this time (see ref. [60] and references therein). Potential candidates for slab break-off SLIPs would be the granite-rhyolite provinces of North America emplaced after supercontinent assembly (ca. 1500–1350 Ma)[1], and the Cretaceous Whitsunday SLIP, where widespread magmatism appeared during continental breakup after the demise of a convergent stage[100].

This study opens new possibilities for the documentation of slab break-off processes in ancient convergent settings. Detection of this process in pre-Cenozoic times is so far limited to the surface geological record, which can be contradictory[106]. Our study documents the oldest case of geophysical constraint on slab-break-off process. We provide an example of how mantle slabs, state-of-the-art plate kinematic reconstructions, and the igneous record can be integrated to better constrain slab break-off episodes back to the late Paleozoic. This approach can be helpful to understand sudden paleogeographic modifications in active margins and the formation and distribution of ore deposits in these settings (e.g., [61,107]). The formation of slab break-off-related ore deposits is well illustrated in our case study by a distinct mineralization event between ~285 and 250 Ma where porphyry, epithermal, polymetallic, and intrusion-related ore deposits developed during the emplacement of the Choiyoi Magmatic Province (see ref. [35] and references therein). Finally, future studies assessing the evolution of the southwestern Pangea margin should test this hypothesis by reproducing the mantle structure beneath this area through integration of plate kinematic data and seismic tomography models into subduction numerical models[108].

## Methods

**Tomotectonic analysis.** For the integration of geology, tomographic slices, and the plate kinematic model we used the Gplates 2.0 software freely available at www.gplates.org[109]. The UU-P07 global tomography model and resolution tests for the São Francisco and Georgia Islands slab walls can be downloaded from www.atlas-of-the-underworld.org[43]. The slab gap in the UU-P07 tomographic model was delineated considering general amplitudes above 0.2% for slab boundaries as suggested by van der Meer et al.[39,43].

**Vote maps analysis.** The vote maps were built with the plotting tools from the submachine portal (https://www.earth.ox.ac.uk/~smachine/cgi/index.php) of Hosseini et al.[110]. To build positive wave speed vote maps we used 26 global models (P-Wave and S-wave) with most of them differing in data selection and parametrization, and regularization of the inversion [GyPSuM-S[111]; DETOX P2 and P3[112]; HMSL-P06 and S06[113]; PRI-P05 and -S05[114]; SPani-P and –S[115]; GAP-P4[116]; LLNL_G3Dv3[117]; Hosseini2016[118]; SEISGLOB1[119]; MITP08[120]; UU-P07[45]; TX2019Slab-P and S[121]; S362ANI + M[122]; S20RTS[123]; S40RTS[124]; SAVANI[125]; SAW642ANb[126]; SEMUCB-WM1[127]; SEMum[128]; TX2011[129]; TX2015[130]; see model details in Table S1]. To build lower mantle vote maps we implemented the standard deviation threshold following Shephard et al.[131]. Also, we built high-velocity votemaps applying a zero threshold (Supplementary Fig. S4). We built these maps at depth between 2800 km and 1800 km at steps of 200 km, corresponding to depths associated with late Paleozoic to Mesozoic subduction zones beneath southwestern Pangea[42,43].

**Geochemical analysis.** As suggested by Hildebrand et al.[60] and Whalen and Hildebrand[61], the geochemical dataset of Upper Carboniferous-Jurassic igneous rocks[132–153] was filtered by $SiO_2$ contents between 55 and 70% and aluminum saturation index (ASI) < 1.1 (filtered $n = 379$) to be plotted in the arc-slab break-off-within-plate tectonomagmatic discrimination diagrams. The filtered dataset and the references are included in Data file S1 (Supplementary information).

To build the tectono-magmatic discrimination diagrams, Hildebrand et al.[60] and Hildebrand and Whalen[61] compared intermediate and acidic I-type granitoids of similar compositions (55–70% $SiO_2$, $Na_2O + K_2O > 1$; ASI < 1.1) using large geochemical data compilations of several active and ancient post-collisional settings. In Whalen and Hildebrand[61], intraplate magmatism was also included in the tectonomagmatic discrimination diagrams. Hildebrand et al.[60], noted that during the development of slab break-off events, the magmas mostly come from the partial melting of a metabasaltic/gabbroic source (i.e., eclogite) compatible with the upper portion of the subducted slab. The latter takes place in addition to variable decompression melting of the sub-slab asthenosphere, which mainly occurs in very shallow slab break-off events[60]. The slab break-off-related magmatism, referred to as slab failure magmatism by the authors, has several distinctive geochemical features that allow it to be distinguished from the arc and A-type magmatism. The slab break-off magmas are enriched in Sr, Nb, Ta, Eu, and depleted in HREE and Y relative to the arc magmatism (see ref. [61] and references therein). These features are produced by the partial melting of an eclogitized slab that leaves a garnet-bearing and plagioclase-free residue, partitioning the HREE into the residual garnet, but not the Sr and Eu due to the absence of plagioclase (see ref. [61] and references therein). For this reason, the slab break-off magmas have high La/Yb (>10), Sm/Yb (2.5), Gd/Yb (>2), and Sr/Y (>20) ratios. Likewise, the instability of rutile or another Ti-rich phase, added to the residual garnet, causes the high Nb/Y (>0.4) and Ta/Yb (>0.3) ratios in the slab break-off rocks[60,61]. Conversely, the arc magmatism is generated by the partial melting of a spinel-plagioclase-bearing source, leaving a residue of pyroxene, plagioclase, and rutile, and hence, the ratios mentioned above are lower (see ref. [61] and references therein). To discriminate the slab break-off magmatism from A-Type granites, Whalen and Hildebrand[61] followed the suggestions of Pearce et al.[154], who indicated that Y + Nb and Yb + Ta are effective to distinguish orogenic from within-plate granitoids. Considering the latter and a geochemical database, Whalen and Hildebrand[61] established the boundary values between these suites (Nb + Y: 60; Ta + Yb: 6).

**Paleogeographic maps.** To build the Late Carboniferous to Early Jurassic paleogeographic maps, we used as a background the plate kinematic reconstruction of Pangea from Matthews et al.[46]. We compiled paleogeographic data with greater detail for the arc, backarc, and intracratonic basins in the study area from Limarino and Spalletti[92] and Limarino et al.[93]. For regions beyond the study area, we used paleogeographic data from Ford and Golonka[155] and Torsvik and Cooker[156]. To run the paleoclimate simulation for the Permian we used the climatearchive.org web-tool, which allows visualizing the results of model simulations for the Phanerozoic from Valdés et al.[95].

## Data availability

All data needed to evaluate the conclusions in the paper are presented in this manuscript and/or the Supplementary information.

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

## Acknowledgements

This study was supported by CONICET and the Universidad Nacional de la Patagonia San Juan Bosco (Grant number: CIUNPAT n°1331 and 1399). We are indebted to Dr. Ross Mitchell and Dr. Suzanne M. Kay for insightful comments to our manuscript and Dr. Kristina Butler and Elaine Keane for English language editing.

## Author contributions

G.M.G. conceived the study. G.M.G. and C.N. wrote the manuscript and built the figures. G.M.G. carried out the tomotectonic and spatiotemporal geochronological analyses, and compiled paleogeographic data. C.N. compiled, analyzed and interpreted the geochemical data.

## Competing interests

The authors declare no competing interests.
