## [Peer Review File · Nature Communications]

REVIEWER COMMENTS

Reviewer #1 (Remarks to the Author):

The basic idea within this manuscript is interesting, the text is well written, and the figures are well constructed. There is potential for a publication of significance, but also more work to be done to make the most important conclusions sufficiently convincing. The new interpretation of a slab failure event may, in itself, be more appropriate for a domain specific journal. To go beyond this, the authors need to make much stronger one of two aspects (or preferably both) - either the link to mass extinctions, or the implications for interpreting SLIPs and their origins throughout Earth history. I expand on these two points in my comments on the discussion text. I also note that some important previous work is not cited - it will be important to more clearly integrate the findings from existing literature into this study.

Specific comments:

A general question about the subduction history:

The text focuses on the oldest subduction we might hope to image from tomography along the South American margin, but nothing is illustrated about what we know of more recent subduction along this same margin. Yet, the younger history should be better constrained, so we could use this to calibrate the optimal sinking rates specific to the history in this region. There are a number of studies that have addressed this, for example Chen et al (Nature 2019) provide one interpretation for the Late Cretaceous to present with some further speculation on how the tomographic images relate to subduction back to ~200 Ma. Perhaps your interpretation would be consistent, perhaps it would differ, but your interpretation would anyway be more convincing if the full history back to the proposed slab failure could be reconciled. This could easily be illustrated on profiles such as those in figure S2 but extended further to the west.

Continuing this point, it is surprising that the recent work of Gianni, Navarrette and Spagnotto (Scientific Reports, 2019) is not mentioned at all in this paper, since that work would seem to contain the authors interpretation for slab dynamics in the same area in the Triassic.

Another important study not cited is that of Nelson and Cottle (Lithosphere, 2019). That study provides evidence that the ChMP extends into West Antarctica (as well as proposing links to global climatic events). How do we reconcile their interpretation of ChMP extending much further with the geographical extent of the slab failure proposed here?

A point about distances: the manuscript refers to the extent of slab gaps and velocity anomalies, but doesn't make clear whether this distance is measured at the Earth's surface or corrected for the changes in Earth's radius (ie., since the velocity anomalies are located deep within the Earth where the planetary radius is much smaller). The implication is that the anomaly labelled on figure 2a as '2,800 to 3,000 km wide' is actually much smaller at depth. Some clearer explanation of this point would be welcome.

On the reconstructions used:

The authors state that they use two different reconstructions, and why they are different. But, what about WHY you use them, why could we expect them to be different? What should we use?

My opinion is that no absolute reconstruction for this time period is without issues, and it is arguably ok to use either of the reconstructions you have. However, there are a few things to bear in mind on how this could be done more robustly.

The region you are studying is on the margin of Gondwana, because of this (and because of how these global reconstruction models are organised as a hierarchy of rotations with Africa at the top), you only really need to consider the different reference frames that tie Africa to the deep Earth. (the other factors in your case would be to consider the way Southern Africa, South America and the Patagonian blocks could be reconstructed differently, but this is a relatively small concern, the reference frame is the main factor). So a global full-plate model is not really needed for what you are studying - and, while I can understand that the Matthews et al and Young et al reconstructions are obvious choices since they are the only global, full tectological models that cover the time period you are interested in, you end up relying on reconstructions

using absolute reference frames that are either based on the assumption of all LIPs arising from the edges of LLSVPs that never move (a rather doubtful assumption, but defensible in the absence of an obviously better alternative) or on paleomag without a True Polar Wander correction (which technically invalidates it for this purpose, despite that Patagonia ends up in the same place as for the other case). Personally, I would start by using the absolute positioning of the continents from van der Meer (2009) at ~250-260 Ma since you know this is calibrated for the entire subduction history imaged by tomography (at least in theory). If this works then we're more confident in your model; if not then this mismatch itself is quite interesting, and you can still make the argument based on the other absolute positioning methods.

With regards to the display of tomography results:

The authors use two very similar colormaps to plot both velocity anomalies and tomography velocity maps. However, I would strongly suggest to change the colormaps for the velocity maps, since the use of a diverging colorscale here is counterintuitive (unlike for the seismic velocities which can be both positive and negative around zero). I realize that the color scheme used is the default on the submachine website and has been used in previous publications, but this does not mean it is a good choice and it is very simple to change it into something more appropriate.

Line 42: Here you refer to 'Slab failure/break off' events. So, these are interchangeable terms for the same thing? If so, then my impression is that the term 'slab break-off' is much more widely used in the literature, so I'd suggest to use that. If a 'slab failure' is not exactly the same as a 'slab break-off', then please explain what the difference is.

Line 59-60, I am not sure what kind of 'independent constraints' this could be referring to. Are you basically saying 'we only have geochemistry, nothing else'? Rephrase the sentence to be more clear.

Paragraph beginning on line 316: This is an important paragraph, since it is where you address the question 'what causes SLIPs?' This needs to be much more developed, specifically:

The only explicit comparison made is to the Sierra Madre Occidental SLIP. What about others? After all, there are not a huge number to review (based on Bryan and Ferrari, 2013, there only seems to be the Whitsundays and the Chon Aike, at least within the time since Pangea assembled - if there are earlier examples then these would be interesting to consider too).

Can we exclude the 'slab failure' model for all these other SLIPs (the final sentence in the abstract presupposes that we know with reasonable certainty what caused all other SLIPs, but since you give no information on the prevailing models for any other SLIPs, the reader has no idea what this assertion is based on)

Even if previous work has not favoured a slab failure scenario for other SLIPs, would it be possible to reinterpret them in view of your new results? I would argue that your results are more significant to a wide audience if you can show that this is not just some unique occurrence specific to one area, but has implications for people studying SLIPs generally, yet you seem to be framing the conclusions as if people studying other SLIPs don't need to care much about the ChMP because it is in any case different from others (which it may be, but I don't see the evidence presented either way).

In summary, what would make this paper of interest to a more general audience would be a more holistic comparison that addresses the question 'what causes SLIPs?' in a more holistic manner (but still centred around the ChMP case).

Paragraph beginning line 332: This is the paragraph dealing with the second potentially important implication of the model, the control of South American slab failure on global mass extinction. The main problem is that the connection is weak and speculative. The middle two sentences of the paragraph give a vague image of how this process could work - this is the beginning of an idea, but the reader is left to do most of the work to understand this idea fully by looking into the cited references. It seems that most of the idea is derived from suggestions in the work of del Rey et al (2019) and Spalletti and Limarno (2017), who develop the link between ChMP and other global events around the Permo-Triassic boundary. So, what is new here? I understand that this study is proposing a different geodynamic mechanism to explain the volcanism, but the ChMP itself is already documented and the link to a mass extinction already proposed. Why is it important for a general audience to know that the volcanism was due to slab failure as opposed to

a different (also subduction-related) process? Perhaps there is some specific detail of the slab-failure model that is better able to explain the way As with the previous paragraph, if there could be some indication of how this could have wider implications for studying other events in Earth history (e.g. a general link between slab failure and magmatic flare-ups? Not necessarily through a specific SLIP).

Reviewer #2 (Remarks to the Author):

The authors used the correlation between the seismic tomographic images of subducted or delaminated lithosphere as an independent constraint to assess the nature and evolution of the silicic Choiyoi Magmatic Province. The seismic boundaries of the São Francisco slab wall and the Georgia Islands anomaly have been used as tie-points to identify the limits of the missing slab record at depths starting from 2200km to the core mantle boundary. For this purpose, they used two different plate kinematic reconstruction models and constant rate of slab sinking for the detached slabs. They illustrated in different scenarios which one fit more to their findings. This study gives a reasonable explanation for the origin of magmatic province and shows even a late Palaeozoic-early Mesozoic seismic anomaly has been preserved in the lower mantle and still can be connected to the surface record via reconstruction models. This research will certainly feed the database for absolute plate motion research. A paragraph about the offset between the present-day location of the Choiyoi Magmatic Province and its related seismic anomaly can also give insight to the readers who are interested in the relative motion between the lithosphere and the average mantle.

In general, the manuscript is clearly written, logically organized, and reasoned well. The methodology is well presented, illustrated and annotated. The methodological steps are systematically presented and illustrated. The problem regarding to the specific areas are clearly defined. Minor modifications to the manuscript would mainly serve to clarify some of the text or figures.

Comments

Line 5: It would be better to emphasize the "major geodynamic change". What kind of change? from ... to ...

Line 35 & 249: Instead of "contractional deformation" "shortening" would be more suitable. The term contraction refers to the shrinking of the consolidated crust of the earth as a consequence of its continuous cooling.

Line 93: Minor lateral migration after the slab breakoff. When the slab is attached, it can be dragged laterally for hundreds of kilometres (Spakman et al. 2018, <https://doi.org/10.1038/s41561-018-0096-6>)

Line 142: Is Figure 1b plotted on a reconstructed paleogeography or is it the present-day configuration?

Line 150: Are there any significant differences between Young et al. and Matthews et al. plate kinematic reconstruction? Please add a sentence or two about the comparison. This can also be discussed in the discussion section.

Line 264: Is there any record on the surface that can support this hypothesis?

Line 284: Please explain further why this scenario is unlikely.

Figure 1: It would be better to use a contrast colour to show study area in Figure 1a.

Figure 2: 2a has two ChMP polygons on it, are these drawn for two different trench locations for 280 and 260 Ma? If so, it would be better to label or colour them separately to avoid the confusion for the ages they represent.

Figure 2e: Why do you think anomaly of the missing slab record in the tomographic images are getting shallower eastward? It reaches 2000 km depth and maybe more. With your calculation this corresponds to 200 Ma.

Fig. S3 is missing in Supplementary file instead Fig. S4 appears twice. I think the Figure caption needs a correction.

Fig. S5: the order of the maps is confusing, it would be better to arrange them in an increasing order or the other way around.

It has been discussed earlier and in different parts of the text but as a conclusion it would be nice to sum up the ideas in one paragraph. What is the type of the slab failure according to your study? How was it originated? Is there any evidence on the surface, for example the spatial distribution of the magmatics along strike and their age propagation? (by referring to Fig. 4)

Reviewer #3 (Remarks to the Author):

Review by Suzanne Mahlburg Kay comments to authors.

This manuscript by Gianni and Navarrete addresses the origin of the silicic Choiyoi magmatic province in South America, which marks a major change in crustal and mantle magma conditions at the Gondwana margin coinciding with the final assembly and initial breakup of the Pangean supercontinent. The topic was addressed some time ago in a series of papers by Kay et al. (1989), Mpodozis and Kay (1990, 1992) and Ramos and Kay (1991), who suggested that the Choiyoi province formed on the South American margin after the major collision of Laurentia with Pangea as subduction ceased along this part of the Gondwana margin with the final terrane collisions leading to slab rupture (break-off) and an associated massive mantle magma upwelling that generated the crustal melting that created the Choiyoi Province (called the ChMP or Choiyoi SLIP in this paper). The Mpodozis and Kay papers argued that important subduction related volcanism did not return to the margin until about the earliest Jurassic marking an important interruption in arc volcanism on or at least on this part of the Gondwanan margin. They supported their hypothesis with the major and trace element and isotopic data available at the time. Subsequently, a series of papers appeared arguing that this was incorrect and that subduction was continuous along the margin throughout this time. Some argued that an eastward sweep in silicic volcanism resulted from steepening of a flat subduction zone. This controversy arose from what Gianni and Navarrete call an "ambiguity" in the geochemical data that comes from analyses plotting in arc fields on discriminant diagrams. Among the problems was the plotting of data with little attention as to the possibility that analyses could reflect extensive melting of older underlying arc crust producing an inherited rather than an active arc signature.

In this vein, it is very refreshing to see this paper by Gianni and Navarrete that uses tomographic evidence to define a possible slab gap that is consistent with slab break-off and a major change in geodynamic conditions at the time of Choiyoi magmatism, and that reopens the use of geochemical data in making a case for continuous subduction during this time. Calling attention to the possible distribution of slabs at the time of the Choiyoi from evidence in the lower mantle is an important advance. The appropriateness of the slab configurations chosen and the concept of a "vote map" for selecting a configuration are outside my expertise. A question is whether the tomographic data used in the vote map analyses here remains the most up-to-date as the status of tomographic data is rapidly evolving as new seismic data are incorporated and processing techniques improve (see comment on line 116).

The re-raising of the question of the interpretation of the geochemical data from the Choiyoi by Gianni and Navarrete is also a useful exercise and their demonstrating that there is an avenue for an interruption in subduction supported by the geochemical data as suggested by Mpodozis and Kay is a step forward. In this regard, the authors refer to a "discovery of a geochemical signature of slab failure" when they plot a modern compilation of geochemical analyses on a series of x-y chemical plots from Hildebrand et al. (2018) and Whalen et al. (2019) that have a field for slab breakoff. The basis as to why these multiple and somewhat repetitive diagrams, which are based on long-established geochemical principles, might work could be better explained so the chemistry does not again become a "black box" as in too many previous papers on the Choiyoi. The diagrams are said to provide the evidence that subduction was not continuous and that slab rupture and crustal thickening events are recorded in the geochemical data. Although the interpretation is very reasonable, there is also a danger to these diagrams, as signatures that plot in the slab breakoff field exist in the Andes today. Julian Pearce, one of the early creators of the diagrams often used for the interpretation of the Choiyoi, once said that his greatest regret in publishing these discrimination diagrams was that they were too often misused by those who did not understand their basis.

Outside of the immediate theme of this paper, the manuscript here does not address whether the Choiyoi event has a place in the supercontinent cycle and the relative importance of subduction on the margins of the Pangean supercontinent. Nevertheless, the argument here is in accord with the Mpodozis and Kay (1992) suggestion that subduction was not prominent along the South American margin at the time of Pangea and could be significant in the ongoing discussion of how continuous or discontinuous subduction was along the entire Pangean margin.

There is a current fad to write papers like text messages despite the fact that limiting abbreviations makes paper easier to follow for the reader. This paper proliferates this trend using SWPM for South Western

Patagonian margin, WPM for Western Patagonian margin, ChMP for Choiyoi magmatic province, SLIP for silicic large ignimbrite province, etc.

Comments keyed to line number.

Line 5 and 6 in abstract. Consider limiting abbreviations that make papers hard to read. Terms like SWPM for southwestern Patagonian margin, SPM for southern Patagonian margin and ChMP for Choiyoi Magmatic province do not add clarity for readers who just have to remember what they mean.

Line 8. Controversy is largely due to casual misuse and lack of understanding of the chemical data rather than the ambiguity of the data.

Line 10. Informs is a strange word. How about "the lower mantle slab record beneath the southwestern Pangean margin that provides clues to late Paleozoic-Mesozoic subducting slab configurations, geochronological information and properly interpreted geochemical data."

Line 15. This is a strange statement - "discovery of a geochemical signature of slab failure". Not a discovery by those who use geochemistry as it should be used rather than just plotting analyses on discriminant diagrams. The whole geochemical data set needs to be considered. A similar case can be made for improper use of seismic tomographic data.

Line 17. Text says findings render the Choiyoi magmatic province the only silicic large igneous province entirely formed by slab failure and the oldest example of a geophysically constrained slab loss event. Although, not geophysically constrained, another could be the Precambrian North America granite-rhyolite province.

Line 27. Kay et al. (1989) used Choiyoi Magmatic Province. The designation ChMP is a recent construct.

Line 33. The Choiyoi province was called a large silicic magmatic province before 2020, but has not always been designated by the abbreviation SLIP.

Line 34-39. The referencing in this section seems rather scattered for the points made.

Line 46. Has similarities to the Mpodozis and Kay (1990, 1992). See summary figure that shows subduction in the pre-Choiyoi, followed by uplift and collision, and subsequent formation of the Choiyoi province after ~ 250 Ma.

Line 48. Moreover, this makes perfect sense with the Mpodozis and Kay (1992) model, which was tied to both geologic and geochemical data.

Line 49 - 52. Many of these papers use the geochemistry in an unsophisticated way leading to difficulties in interpretation and lumping analyses on plots without much attention to what the samples are or where they are from.

Line 55. The isotopic data fit well with the slab break off model of Mpodozis and Kay, who made a point of a temporal isotopic shift based on the isotopic data in their paper and that available in 1992. The isotopic shift was correlated with slab break-off and a switch from a subduction to a post subduction extensional regime. This agrees well with more recent isotopic data used to indicate a shift from a source with a more crustal influence to a larger mantle component in the Choiyoi magmas.

Line 59. Better not to stress the ambiguity of geochemical data - the problem is how the data are used and interpreted - not the actual data.

Line 70-72. And this is how Mpodozis and Kay (1992) interpreted the geochemical data before all became confused on classification diagrams that did not consider the genesis of the magmas. The diagrams here are

somewhat repetitious as Y+Nb denote a similar behavior to Ta+Yb, Y+Nb is similar to Ta+Yb, and low Yb contents and high Sr/Y both indicate depletion of heavy REEs, etc. This is the same chemistry that has been used to indicate garnet retention, arc signatures and slab breakoff for years.

Line 74-76. Needed? "Finally, this study illustrates an interdisciplinary approach to solve geological problems testing geodynamic hypotheses in ancient convergent settings back to late Paleozoic time."

Line 92. The term tomotectonics seems to have come into the literature in ~2017 and has caught on since. A minor point, but this term is not used in the references cited here.

Lines 116 and 128. Please briefly explain what you mean by a vote map and vote map stacking, as many readers will not be familiar with this methodology. Maybe you could move the discussion on vote maps from the methods section on lines 365 to 172 to this point in the paper.

There are a series of recent papers on tomographic models looking at slabs at depths of 200 to 1200 km using P-wave tomography [Portner et al. 2019, JGR & Mohammadzakeriet et al 2020, JGR] and S-wave tomography (Rodriquez et al. 2021, Geophys. J. Int)]. How well do these tomographic models match and mesh with the P and S wave tomographic models at greater depths featured here? Particularly, could any these of these older lower mantle models be affected by the more recent increase in seismic tomographic data available for South America? Just a question.

Line 120. Data base in file S1 used here is from references 102-123 and is a diverse data set from a wide area. You could add a few more comments in the text as to what is included in this data set and why. You do not reference the much smaller 'less complete' data set included in and compiled by Kay and Mpodozis (1992) from which they arrive at generally similar conclusions.

Line 131. Tomographic data are from references numbered 39 and 40 - Van der Meer et al. (2012, 2018). Maybe you could cite the author's names as the Nature referencing system puts the burden on the reader to search for the reference by number.

Line 135. Same comment as 131. Say that data is from Spikings et al. (2015) and Chew et al. (2016).

Line 142. Text says reconstructions in Fig.1a, b with tomographic depths. Should this be Figure 2?

Line 159. Note partial melts of preexisting arc rocks can lead to arc signatures, but this does not necessarily signify these magmas formed in an active arc environment. This problem with plotting silicic rocks on discrimination diagrams has long plagued the interpretation of the Choiyoi. Why show plots from both Hildebrand et al. and Whalen and Hildebrand when you do not discuss why these specific elements are plotted. The first set is largely based on the HFSE and the second on the slope of the REE partners. Slab failure leads to depleting the heavy REE as indicated by low Yb and Y (depleted HREE), high Gd/Yb ratio and elevated Sr contents - all of which indicate a garnet signature with a breakdown of plagioclase. Mpodozis and Kay (1992) pointed out these signatures in their Cochiguas and El Vulcan samples. Note also that the decrease in $87\text{Sr}/86\text{Sr}$ in Fig 3 of Kay and Mpodozis indicates a less enriched isotopic source after the Cochiguas and El Vulcan Units.

Line 163-165. La/Yb versus Yb diagrams - back to adakites and the name that means little more than an undefined high Sr content and a low HREE content indicated by Yb or Y that together imply garnet retention and feldspar breakdown. As a comment, most of these adakites look little like the Adak lava in the Aleutians from which Drummond and Defant took the name. A question - what at the nature of the samples that plot with La/Yb ratios near 100 on Figure S8

Fig. 3 - Line 803. Note typo - should be 261-246 Ma, not 61-246 Ma.

Line 176. Eastward broadening of Choiyoi province - up to 500 km from current trench, much is actually in today's forearc. What does this mean considering strike slip and the possible effects of forearc subduction erosion?

Line 185. How clear is this arc signature and how extensive is it in late Triassic and Early Jurassic igneous rocks? Silicic rocks can have significant inherited crustal components whose presence do not confirm they erupted in an active arc. I have not read the papers recently, but I don't recall overwhelming cases for extensive arc rocks in Choiyoi magmas proper in papers denoted by references 23 (Oliveros et al.), 29 (Lopes de Luchi et al.) or 59 (Vasquez et al.). Figure S10 gives no indication of silica content or evolutionary stage of the plotted samples and case needs to be made based on widespread mafic samples.

Line 207. This is pretty similar to the slab breakoff hypothesis in Mpodozis and Kay (1992). The difference is that you restart the arc sooner – but what is the real evidence in the geochemistry as to when normal arc volcanism reappears.

Line 210. Please review the evidence that Chernicoff et al. offered for removing the Intracratonic magmatic belt from the Choiyoi magmatic province. How is Figure 3 relevant to this argument?

Near line 215. The steepening of a shallow subduction zone was always a difficult model to visualize particularly when an analogy was made with the modern Puna and ignimbrite flare-up, which is a very different story from the Choiyoi.

Line 217. Again, Mpodozis and Kay (1992) argued against this scenario, called for slab break off and a break in subduction with the evolution of the Choiyoi Province being closely linked with events related to Pangea. Do you think there is any connection with the Pangean supercontinent or is the Choiyoi province just an unrelated episode of slab break off that happens to occur on the Pangean margin?

Line 227-243. In other words melting of eclogite. How about a component from partial melting of the lower crust by mantle-derived magmas? Again, this is more or less what Mpodozis and Kay argued – they did not mention high-K mafic magmas. The huge improvement over their model is your perspective on the slab gap based on evidence from the lower mantle.

Line 245-275. Discussion of Southern and Northern Patagonian terrane accretion. This section is hard to follow for those not familiar with the latest on this ever changing and seemingly never ending debate. Need more context for the reader to follow this.

Line 258. Autochthonous in what time frame?

Line 259. Is all of this discussion based on analyses falling in a field labeled slab failure? How do you explain the paleomag data? The discussion of the North Patagonian Terrane needs more context.

Line 275. There has been too much emphasis on the details of the Equis terrane as a means to detract from the Kay and Mpodozis (1992) proposal for the termination of subduction with slab break-off. The slab failure event was never said to be restricted to the region of the Equis terrane. A way was needed to stop subduction and a possibility was to do this with terrane collisions in the final amalgamation of Pangea. The Equis terrane was proposed to end subduction along part of the south central Andes (and explain the San Rafael orogeny) and was fashioned on a proposal in Australia. Although speculative, all of the evidence for the Equis and similar terranes could be absent due to subsequent forearc subduction erosion. How is lack of concrete evidence for a hypothetical X terrane (equis in Spanish) an argument against the slab break-off model of Mpodozis and Kay? What is the alternative evidence for subduction of an ocean ridge and how does this work with the San Rafael event? Seems more tenuous than a hypothetical terrane removed by forearc subduction erosion. The whole issue leads to the larger question of whether or not there was subduction along the margin of Pangea existed and the influence of supercontinents. The seismic evidence featured here could be key in examining this issue.

Lines 291-298. Accretionary fragments associated with the end of subduction along the margin of the Pangean super continent makes some sense and could include small terranes with the speculative Equis terrane being one of these fragments. Need to take into account that these are final events in the formation

of Pangea and the beginning of the incipient break-up and that this is a special time in Earth history.

Line 304. This discussion more or less supports the concept of a series of "Equis" terranes.

Line 319-321. There was also early discussion of this in comparing the Southern America Choiyoi-Chon Aike provinces with the North American granite-rhyolite province in Kay et al. (1989) and Mpodozis and Kay (1990, 1992). The point was that massive continental rhyolite provinces are found in relatively young crust and are likely critical in producing stable crust- they are not common in older continental crust.

Line 328. If you read through the lines, slab failure is also suggested from Peru to Australia in the Mpodozis and Kay (1990, 1992) papers. What specifically are you referring to when you mention enormous amount of andesitic rocks in the lower portion of the Choiyoi magmatic province. Is this really pre-Choiyoi Paleozoic subduction and a confusion on how the Choiyoi province is defined?

Line 380. Why is this discussion put at this point in the paper? It would be better if it was where the topic is highlighted in the paper.

Line 741. Reference 1122. Note Rapela is repeated.

Figures.

Fig 1. Could write out names for AR, IMP and ET (sounds like the extra-terrestrial in the movie ET) - there is room on the figure. The Equis terrane proposed by Mpodozis and Kay actually went further south than shown as it was largely based on a transect at 29° to 32°S.

Figure 4. Careful of what plotting ages of silicic rocks at the current distance to the trench actually means. What does this really mean as there has been a lot of strike-slip motion and possible forearc subduction erosion along the margin since the Choiyoi. As the modern volcanic line is ~ 300 km from the trench, many of the localities are in the modern forearc and only a small portion in the modern back-arc.

Reviewer #1 (Remarks to the Author):

The basic idea within this manuscript is interesting, the text is well written, and the figures are well constructed. There is potential for a publication of significance, but also more work to be done to make the most important conclusions sufficiently convincing. The new interpretation of a slab failure event may, in itself, be more appropriate for a domain specific journal. To go beyond this, the authors need to make much stronger one of two aspects (or preferably both) - either the link to mass extinctions, or the implications for interpreting SLIPs and their origins throughout Earth history. I expand on these two points in my comments on the discussion text. I also note that some important previous work is not cited - it will be important to more clearly integrate the findings from existing literature into this study.

Gianni and Navarrete: We are thankful to reviewer 1 for this insightful and detailed revision. We have followed the main suggestions regarding exploiting our findings and their implication for interpreting SLIPs and their origins. Also, we have further clarified our idea connecting magmatic migration linked to the slab break-off event and Permian climate change and extinction. We have followed all the additional comments. Detailed responses to the comments above and modified line numbers of our manuscript are found below, in answers to specific comments by reviewer 1.

Specific comments:

Reviewer 1: A general question about the subduction history: The text focuses on the oldest subduction we might hope to image from tomography along the South American margin, but nothing is illustrated about what we know of more recent subduction along this same margin. Yet, the younger history should be better constrained, so we could use this to calibrate the optimal sinking rates specific to the history in this region. There are a number of studies that have addressed this, for example Chen et al (Nature 2019) provide one interpretation for the Late Cretaceous to present with some further speculation on how the tomographic images relate to subduction back to ~200 Ma. Perhaps your interpretation would be consistent, perhaps it would differ, but your interpretation would anyway be more convincing if the full history back to the proposed slab failure could be reconciled. This could easily be illustrated on profiles such as those in figure S2 but extended further to the west.

Gianni and Navarrete: The reviewer is right. These concerns are in part caused by missing references and additional explanations necessary in our previous manuscript. Tomotectonic reconstructions dealing with the younger subduction evolution from Mesozoic to Cenozoic times have already been addressed in great detail in previous papers (Chen et al., 2019, Nature; Gianni et al., 2019 Sci. Rep.; Mohammadzaheri et al., 2021 JGR). In this regard, our sensitivity analysis, including average slab sinking rates ranging from 1-1,3 cm/yr, already covers the same lower

mantle slab sinking rates used by those studies. The latter is meaningful because these values are the ones that most effectively reproduce the surface igneous and sedimentary records of the younger subduction history (1 cm/yr, Gianni et al., 2019; ~1,3 cm/yr, Chen et al., 2019; ~1-1,1 cm/yr Mohammadzaheri et al., 2021). Noteworthy, the most recent study of Mohammadzaheri et al. (2021) found that the most appropriate sinking rates for Meso-Cenozoic slabs beneath South America are 1-1,1 cm/yr, which are the same values that best reproduce the surface records of slab-break-off (ChMP) and arc magmatism in our study area (Fig. 2 and S3). In Gianni et al. (2019), we found the same results when analyzing the latest Triassic to Cretaceous subduction. Also, a recent study that inspected the mantle structure corresponding to the Cretaceous to present subduction of the Caribbean region found similar optimal slab sinking rates (0,8-1 cm/yr, Braszus et al., 2021, Nat Comm.). Therefore, our reconstructions already consider the optimal sinking rates specific to the subduction history of South America and hence, are consistent with the younger subduction evolution of this region. As suggested by the reviewer, we now refer to the latter studies and the above-stated points to highlight the suitability of the chosen slab sinking rates for the sensitivity analysis (see new lines 118-121 and 233-235). Also, we added the tomographic profile suggested by the reviewer in Fig. S2 (profile V-VI). We show a profile crossing the present trench in northern South America that illustrates the mantle structure associated with Permian to Present subducted slabs and indicated the optimal slab sinking rates determined for the slabs in that region (Chen et al., 2019; Nature; Mohammadzaheri et al., 2021 JGR; van der Meer et al., 2018; This study).

References:

Mohammadzaheri, A., Sigloch, K., Hosseini, K., & Mihalynuk, M. G. (2021). Subducted Lithosphere under South America from Multi-frequency P-wave Tomography. *Journal of Geophysical Research: Solid Earth*, 126, e2020JB020704.

Braszus, Benedikt, et al. Subduction history of the Caribbean from upper-mantle seismic imaging and plate reconstruction. *Nature communications*, 2021, vol. 12, no 1, p. 1-14.

Reviewer 1: Continuing this point, it is surprising that the recent work of Gianni, Navarrete and Spagnotto (Scientific Reports, 2019) is not mentioned at all in this paper, since that work would seem to contain the authors interpretation for slab dynamics in the same area in the Triassic. Another important study not cited is that of Nelson and Cottle (Lithosphere, 2019). That study provides evidence that the ChMP extends into West Antarctica (as well as proposing links to global climatic events). How do we reconcile their interpretation of ChMP extending much further with the geographical extent of the slab failure proposed here?

Gianni and Navarrete: The reviewer is right. We now refer to Gianni et al. (2019) in lines 95, 100, 107, 114, 127, and 235.

Regarding the work of Nelson and Cottle (2019), these authors conclude the following: New zircon U-Pb and Hf isotopic data for Permian detrital zircon in three samples from volcanoclastic sedimentary rocks in central Antarctica record a major episode of explosive continental arc

volcanism at ca. 268 Ma. Hf isotopes indicate that Permian volcanism in central Antarctica is consistent with derivation from temporally and geochemically correlative rocks in South America, Antarctic Peninsula, and Thurston Island, that likely represent a continental arc flare-up related to the Choiyoi Province. Central Antarctica, therefore represents the southernmost documented extension of this broad volcanic and magmatic province that is distinct from continental arc activity recorded in central TAM, Marie Byrd Land, Zealandia, and Australia.

Although regional analyzes of geological databases as this are useful and necessary to understand the geodynamic context of magmatism along the active margins, the study by Nelson and Cottle (2019) appear to us oversimplified in the following aspects:

- 1) This study starts from the assumption that the ChMP was formed in an arc environment (hypothesis of uninterrupted subduction setting) and overlooks important bibliography and a major discussion on the origin of the ChMP. By assuming that all magmatism is of arc nature, these authors can group all magmatism of similar age and similar magmatic sources into a single subduction-related magmatic event extending the ChMP to the south up to Central Antarctica. However, in their regional analysis, these authors overlooked a more in-depth discussion in the literature about the origin of ChMP. The geodynamic context of the ChMP is part of an intense discussion with varying interpretations, which are: i) uninterrupted subduction setting (Kay et al., 1989; Llambias et al., 1995; Franzeze and Spalletti, 2001; Mpodozis and Kay, 1992; Pankhurst et al., 2006; García-Sansegunado et al., 2014), ii) subduction interrupted by slab break-off (Brown et al., 1991; Strazzere et al., 2006; del Rey et al., 2014; 2019; Varela et al., 2015; Coloma et al., 2017; Bastías-Mercado et al., 2020; Gregori et al., 2020; Oliveros et al., 2020), and iii) a combination of both previous hypotheses, subduction up to ~260 Ma and subsequent slab break-off (Kleiman et al., 2009; Rocher et al., 2015; Ramos et al., 2020; Martina et al., 2020; Martínez Dopico et al., 2019). The choice of the subduction-related hypothesis is somehow arbitrary, and the regional-scale genetic correlations made by these authors for the Permian magmatism of southwestern Gondwana to us are debatable.
- 2) In the light of the existing literature from the North Patagonian massif, the interpretation of a continuous arc from Peru to Australia is doubtful. Previous studies in the North Patagonian Massif coincide in the interruption of the subduction process sometime between Permian to Early Triassic, but these mainly disagree on the causes of this process and the exact timing. The magmatism associated with ChMP in Central and North Patagonia (North Patagonian massif) is variably interpreted as related to the collision of the Nord-Patagonian massif terrane (Rapallini et al., 2005, *Geol. Soc. Lond.*; Ramos, 2008 *J. S. Am. Sci.*; Martínez-Dopico et al., 2020, *J. S. Am. Sci.*; Ramos et al., 2020, *Ameghiniana*; among others); and/or the Deseado massif (southern Patagonia terrane), where the two strongest hypotheses, supported by geophysical, field structural, and geochemical data point to entirely post-collisional magmatism since ~290-280 Ma (Pankhurst et al., 2006 *Earth Sci. Rev.*; Fanning et al., 2011, *J. S. Am. Earth Sci.*) or subduction magmatism that

becomes post-collisional at 260 Ma (Martinez-Dopico et al., 2019, *J. S. Am. Earth Sci.*; Lopez de Lucci et al., 201, *J. S. Am. Earth Sci.*; Ramos, 2008, *J. S. Am. Earth Sci.*). These collisional events were omitted in the analysis by Nelson and Cottle (2019) and are not compatible with the subduction scenario from Peru to Australia proposed in the final scheme of that study.

On the other hand, the totality of the previous studies in southern South America and the Antarctic Peninsula has shown that age equivalents of the ChMP are compatible with the existence of a magmatic arc and active subduction in this region (e.g., Castillo et al., 2016, *Gondwana Res.*; Fanning et al., 2011 *J. S. Am. Earth Sci.*; Elliot et al., 2016, 2017; van der Meer et al., 2018, *Tectonophysics*; Bastías et al., 2020, *Lithos* and references therein). The latter interpretation has been independently supported by previous tomotectonic studies (van der Meer et al., 2010, *Nature Geosci.*; 2018, *Tectonophysics*) and also our study (see Fig. 2), indicating the presence of a Permian oceanic slab beneath this region.

- 3) Nelson and Cottle (2019) suggest that shared isotopic signatures between Permian magmatism in South America and Antarctica provide strong evidence for a lithospheric mantle source for contemporaneous Permian magmatism in South America, including the Choiyoi Province, and are consistent with the Choiyoi Province or temporally and geochemically correlative arc magmatism as the source for Permian zircon from central Antarctica. According to those authors, the latter provides support for extending the ChMP into the Antarctic territory. Nevertheless, contemporaneous rocks separated thousands of kilometers, but sharing a similar magma source, do not necessarily ensure the same geodynamic setting. Similar isotopic signatures only indicate rocks formed by similar magma sources. Lithospheric mantle and the continental crust can be involved in many geodynamic settings from active margins under contraction to backarc and intraplate rifts to collisional/post-collisional settings. In support of this argument is the fact that the enriched isotopic signature is shared by the post-collisional ChMP in the North Patagonian massif (Pankhurst et al., 2006; Fanning et al., 2011; Castillo et al., 2017; among others) and the Permian magmatism in southern South America and Peninsula Antarctica, where there is a consensus of a subduction-related setting in this region supported by geochemistry and seismic tomography data (e.g., Castillo et al., 2016, *Gondwana Res.*; Fanning et al., 2011 *J. of South Am. Earth Sci.*; Elliot et al., 2016, 2017, van der Meer et al., 2010; 2018; Bastías et al., 2020, *Lithos* and references therein). Again, we value this effort by Nelson and Cottle (2019), and this type of regional approach is necessary for regions that are still poorly understood. However, the limitation on their regional analysis has to be flagged.
- 4) Another striking point not discussed or explained in Nelson and Cottle (2019) is the major differences in the areal extent of the Permian magmatism in southern South America (Tierra del Fuego)-Antarctic segment and the central Patagonia-northern Chile segment. The ChMP from northern Chile to the central Patagonian sector is characterized by

widespread volcanic/plutonic outcrops covering 909,250 km² and extending in parts to the intraplate area, which is atypical for arc magmatism. The latter leads to the SLIP interpretation for the Permian-Triassic magmatism in this region (see Bastías-Mercado et al., 2020; J. S. Am. Earth Sci.). Whereas, Permian magmatism to the south forms a narrow belt parallel to the plate margin with sparse occurrences of granites in the crustal blocks of West Antarctica (e.g., Antarctic Peninsula, Marie Byrd Land, Thurston Island; Fig. 2) (e.g., Pankhurst et al., 1998; Castillo et al., 2016; Fanning et al., 2011; Elliot et al., 2016, 2017). How are these substantial differences explained in the common subduction context suggested by Nelson and Cottle (2019)? Our hypothesis involving a massive slab loss process in the area classically considered for the ChMP (Sato et al., 2015; Bastías-Mercado et al., 2020, J. S. Am. Earth Sci.) provides a straightforward answer for this volumetric difference in magmatic activity. Additionally, the documentation of Permian slabs in tomotectonic analyses (van der Meer et al., 2010; 2018; and our study) supports previous studies indicating a subduction-related setting for the Permian magmatism in southern South America (Tierra del Fuego)-Antarctic segment (e.g. Pankhurst et al., 1998, J. Geophys. Res. Solid Earth)

It is not our intention to dismiss this valuable work but to point out what we consider limitations in the regional analysis. We followed the reviewer's suggestion about considering this study and introduced additional sentences discussing this work briefly in lines 248-253. Also, we added this work on lines that describe the connection between the end-Permian mass extinction and the ChMP.

Reviewer 1: A point about distances: the manuscript refers to the extent of slab gaps and velocity anomalies, but doesn't make clear whether this distance is measured at the Earth's surface or corrected for the changes in Earth's radius (ie., since the velocity anomalies are located deep within the Earth where the planetary radius is much smaller). The implication is that the anomaly labeled on figure 2a as '2,800 to 3,000 km wide' is actually much smaller at depth. Some clearer explanation of this point would be welcome.

Gianni and Navarrete: The reviewer is right; we need to further clarify this point. We have done so in lines 173-178 by introducing the following lines:

...Cross-sections indicate a lateral extent for this discontinuity of ca. 2,000 km (Fig. 2e). The reconstructed size along these cross-sections, achieved by a surface projection of the lower mantle velocity discontinuity in our tomotectonic reconstructions, indicates an originally larger size of ca. 2,800-3,000-km (Fig. 2a). In any case, the high-velocity discontinuity coincides with the reconstructed position of the Choiyoi Magmatic Province (Figs. 2a,b).

The slab gap distance is measured on the surface after restoration to the surface by Gplates. The slab gap is smaller being ~2000 km in cross-sections on Fig. 2e and larger further east as seen in horizontal tomographic slices in Fig. 2a and b. As two points at the surface in a trench position would get progressively closer during subduction in the deep mantle, projection to the surface in tomotectonic reconstructions would restore this distance. A discussion on possible modifications of the slab gap during deep mantle subduction is provided in lines 238-241 in the discussion section.

Reviewer 1: On the reconstructions used:

The authors state that they use two different reconstructions, and why they are different. But, what about WHY you use them, why could we expect them to be different? What should we use? My opinion is that no absolute reconstruction for this time period is without issues, and it is arguably ok to use either of the reconstructions you have. However, there are a few things to bear in mind on how this could be done more robustly. The region you are studying is on the margin of Gondwana, because of this (and because of how these global reconstruction models are organised as a hierarchy of rotations with Africa at the top), you only really need to consider the different reference frames that tie Africa to the deep Earth. (the other factors in your case would be to consider the way Southern Africa, South America and the Patagonian blocks could be reconstructed differently, but this is a relatively small concern, the reference frame is the main factor). So a global full-plate model is not really needed for what you are studying - and, while I can understand that the Matthews et al and Young et al reconstructions are obvious choices since they are the only global, full topological models that cover the time period you are interested in, you end up relying on reconstructions using absolute reference frames that are either based on the assumption of all LIPs arising from the edges of LLSVPs that never move (a rather doubtful assumption, but defensible in the absence of an obviously better alternative) or on paleomag without a True Polar Wander correction (which technically invalidates it for this purpose, despite that Patagonia ends up in the same place as for the other case). Personally, I would start by using the absolute positioning of the continents from van der Meer (2009) at ~250-260 Ma since you know this is calibrated for the entire subduction history imaged by tomography (at least in theory). If this works then we're more confident in your model; if not then this mismatch itself is quite interesting, and you can still make the argument based on the other absolute positioning methods.

Gianni and Navarrete: The reviewer is right. We have implemented two additional reconstructions in the new Fig. S1 including two different models that constrain paleolongitude back to Paleozoic times. One is the lower mantle slab model of van der Meer et al. (2010), suggested by the reviewer, and the other is the orthoversion model of the supercontinent cycle of Mitchell et al. (2012). As can be seen from our new Fig. S1, during the Permian stage when the ChMP was emplaced, all models indicate a paleogeographic coincidence between the reconstructed margin of Pangea, the Choiyoi magmatic Province, and the slab gap, making this spatio-temporal relation robust. Only the method of Mitchell et al. (2012) shows a departure from the mantle structure but in Triassic times.

Reviewer 1: With regards to the display of tomography results: The authors use two very similar colormaps to plot both velocity anomalies and tomography vote maps. However, I would strongly suggest to change the colormaps for the votemap plots, since the use of a diverging colorscale here is counterintuitive (unlike for the seismic velocities which can be both positive and negative around zero). I realize that the color scheme used is the default on the

submachine website and has been used in previous publications, but this does not mean it is a good choice and it is very simple to change it into something more appropriate.

Gianni and Navarrete: The reviewer is right. We have modified the colorscale of votemaps on Fig.2d and S4 as suggested.

6-Line 42: Here you refer to 'Slab failure/break off' events. So, these are interchangeable terms for the same thing? If so, then my impression is that the term 'slab break-off' is much more widely used in the literature, so I'd suggest to use that. If a 'slab failure' is not exactly the same as a 'slab break-off', then please explain what the difference is.

Gianni and Navarrete: The reviewer is right. These terms are interchangeable. We now use the term slab break-off throughout the text that is the most common way to refer to horizontal slab detachment processes.

Reviewer 1: Line 59-60, I am not sure what kind of 'independent constraints' this could be referring to. Are you basically saying 'we only have geochemistry, nothing else'? Rephrase the sentence to be more clear.

Gianni and Navarrete: This sentence introduced confusion and we decided to remove it following suggestions of Reviewer 3. In the following sentences (new lines 63-66), we indicate that seismic tomography could be used as an independent constrain to assess the competing hypothesis (subduction vs. post-subduction setting).

Reviewer 1: Paragraph beginning on line 316: This is an important paragraph, since it is where you address the question 'what causes SLIPs?' This needs to be much more developed, specifically:

The only explicit comparison made is to the Sierra Madre Occidental SLIP. What about others? After all, there are not a huge number to review (based on Bryan and Ferrari, 2013, there only seems to be the Whitsundays and the Chon Aike, at least within the time since Pangea assembled - if there are earlier example then these would be interesting to consider too).

Can we exclude the 'slab failure' model for all these other SLIPs (the final sentence in the abstract presupposes that we know with reasonable certainty what caused all other SLIPS, but since you give no information on the prevailing models for any other SLIPs, the reader has no idea what this assertion is based on) Even if previous work has not favoured a slab failure scenario for other SLIPs, would it be possible to reinterpret them in view of your new results? I would argue that your results are more significant to a wide audience if you can show that this is not just some unique occurrence specific to one area, but has implications for people studying SLIPs generally, yet you seem to be framing

the conclusions as if people studying other SLIPs don't need to care much about the ChMP because it is in any case different from others (which it may be, but I don't see the evidence presented either way).

In summary, what would make this paper of interest to a more general audience would be a more holistic comparison that addresses the question 'what causes SLIPs?' in a more holistic manner (but still centred around the ChMP case).

Gianni and Navarrete: The reviewer is right. This comment has certainly allowed us to expand the implications of our results (see new lines 420-439). Now we highlight that our study provides new means to unravel the causes of SLIPs, which is part of a long-standing debate (e.g., Bryan et al., 2002; Bryan and Ferrari, 2013; Ernst, 2019). In this regard, tomotectonic analysis is an unappreciated tool that would help to assess the origin of SLIPs worldwide back to Permian times. The methodology followed in our study will be useful to shed light on the often opposed views suggested for the origin of SLIPs (i.e. subduction vs. post-subduction settings), a task that is difficult to assess only from a geochemistry point of view (e.g., the Jurassic Chon Aike SLIP: Plume-related rifting model, Pankhurst et al., 2000 J. Petrol. vs. subduction-related model, Bastías et al. 2021, Lithos). As the highly evolved rocks, characteristics of SLIPs, often do not allow an easy discrimination between original subduction-related signatures from those inherited resulting from partial melting of an arc basement (Brian et al., 2002; 2007), we suggest that tomotectonic analysis indicating the presence/absence of ancient slabs beneath reconstructed margins can provide an independent constraint to assess the cause of SLIPs.

Regarding this comment: The only explicit comparison made is to the Sierra Madre Occidental SLIP. What about others?

We have now included a brief mention to other SLIPs and their proposed geodynamic settings as suggested by the reviewer (see new lines 424-432). Also, we refer to a possible Precambrian analog of the Choiyoi magmatic province in new lines 453-454.

Concerning the question, if can we exclude the 'slab failure' model for all these other SLIPs. We now indicate the following in new lines 448-454:

...Although we highlight slab break-off processes in the origin of the Choiyoi magmatic province, a similar role for other SLIPs remains to be demonstrated in future studies. Potential candidates for slab break-off igneous provinces would be the Cretaceous Whitsunday SLIP, where widespread magmatism appeared suddenly after the demise of a convergent stage (Bryan et al., 2002; 2007), and the granite-rhyolite provinces of North America emplaced after supercontinent assembly (ca. 1500-1350 Ma) (Kay et al., 1989)...

To provide an organized description of the implications of our study, we have reordered the discussion section as follows: First, we assess the implications of our results for the origin of the Choiyoi Magmatic province, the Patagonian geology, and the end-Permian climate and change. Then, we assess the broader implications of our study. There, we highlight the utility of our approach to solving the debated geodynamic setting of SLIPs (convergent vs. non-convergent settings), and then, how this methodology can also be applied to resolve slab-break-off processes back to Permian times. We insist that the latter constitutes a major advance in our field as slab break-off events can so far be geophysically constrained back to Cenozoic times. As indicated in the text, detection of slab break-off events has an exploratory interest as these events are associated with different mineral deposits linked to convergent settings (e.g., Whalen and Hildebran, 2019 and references therein). After this valuable correction, we are confident that the multiple implications of our study will make it of interest to a broad readership.

Reviewer 1: Paragraph beginning line 332: This is the paragraph dealing with the second potentially important implication of the model, the control of South American slab failure on global mass extinction. The main problem is that the connection is weak and speculative. The middle two sentences of the paragraph give a vague image of how this process could work - this is the beginning of an idea, but the reader is left to do most of the work to understand this idea fully by looking into the cited references. It seems that most of the idea is derived from suggestions in the work of del Rey et al (2019) and Spalletti and Limarno (2017), who develop the link between ChMP and other global events around the Permo-Triassic boundary. So, what is new here? I understand that this study is proposing a different geodynamic mechanism to explain the volcanism, but the ChMP itself is already documented and the link to a mass extinction already proposed. Why is it important for a general audience to know that the volcanism was due to slab failure as opposed to a different (also subduction-related) process? Perhaps there is some specific detail of the slab-failure model that is better able to explain the way as with the previous paragraph, if there could be some indication of how this could have wider implications for studying other events in Earth history (e.g. a general link between slab failure and magmatic flare-ups? Not necessarily through a specific SLIP).

Gianni and Navarrete: The reviewer is right. As pointed out by reviewer 1 our idea was not properly developed. We have clarified our idea and expanded our contribution to this topic in new lines 402-419. As indicated in the new version of the manuscript, the connection between the Choiyoi magmatic intrusion, gas release, and climate change has been previously established. Nevertheless, the role of magmatic migration and broadening in triggering this process has been largely overlooked so far. Previous studies in ancient convergent margins have proposed slab break-off as a viable process to drive synextensional inland migration of voluminous magmatic activity (e.g. Decker et al. 2017, J. Petrol.). We suggest that the ~500-900 km magmatic broadening triggered by the slab loss stage documented in our study was a key process allowing distal magma intrusion in organic-rich Paleozoic basins inland southwest Pangea (Figs. 1b and 4). Otherwise, magmatic intrusion during typical subduction-related stages of Andean magmatism (e.g. Oliveros

et al., 2020, Gondwana Res.; Vazquez et al., 2011, Geol. J. Lond.) would have been more progressive, and hence, the impact on climate change, likely weaker.

Reviewer #2 (Remarks to the Author):

The authors used the correlation between the seismic tomographic images of subducted or delaminated lithosphere as an independent constraint to assess the nature and evolution of the silicic Choiyoi Magmatic Province. The seismic boundaries of the São Francisco slab wall and the Georgia Islands anomaly have been used as tie-points to identify the limits of the missing slab record at depths starting from 2200km to the core mantle boundary. For this purpose, they used two different plate kinematic reconstruction models and constant rate of slab sinking for the detached slabs. They illustrated in different scenarios which one fit more to their findings. This study gives a reasonable explanation for the origin of magmatic province and shows even a late Palaeozoic-early Mesozoic seismic anomaly has been preserved in the lower mantle and still can be connected to the surface record via reconstruction models. This research will certainly feed the database for absolute plate motion research. A paragraph about the offset between the present-day location of the Choiyoi Magmatic Province and its related seismic anomaly can also give insight to the readers who are interested in the relative motion between the lithosphere and the average mantle.

In general, the manuscript is clearly written, logically organized, and reasoned well. The methodology is well presented, illustrated and annotated. The methodological steps are systematically presented and illustrated. The problem regarding to the specific areas are clearly defined. Minor modifications to the manuscript would mainly serve to clarify some of the text or figures.

Comments

Line 5: It would be better to emphasize the “major geodynamic change”. What kind of change? from ... to ...

Gianni and Navarrete: We decided to modify this first sentence of the abstract. We now emphasize that what has intrigued geoscientists is the origin of the large volumes of silicic magmatism associated with the Choiyoi magmatic province.

Line 35 & 249: Instead of “contractional deformation” “shortening” would be more suitable. The term contraction refers to the shrinking of the consolidated crust of the earth as a consequence of its continuous cooling.

Gianni and Navarrete: The reviewer is right. This was corrected.

Line 93: Minor lateral migration after the slab breakoff. When the slab is attached, it can be dragged laterally for hundreds of kilometres (Spakman et al. 2018, <https://doi.org/10.1038/s41561-018-0096-6>)

Gianni and Navarrete: The reviewer is right. We have now included a reference to this study and a short comment about this issue in lines 97-98.

Line 142: Is Figure 1b plotted on a reconstructed paleogeography or is it the present-day configuration?

Gianni and Navarrete: It is the present day. We now clarify this point in the figure caption.

Line 150: Are there any significant differences between Young et al. and Matthews et al. plate kinematic reconstruction? Please add a sentence or two about the comparison. This can also be discussed in the discussion section.

Gianni and Navarrete: The reviewer is right. We have now included this discussion in lines 182-187. Also, we have expanded this analysis by including two additional reconstructions with alternative reference frames that constrain paleolongitude (van der Meer et al., 2010; Mitchell et al., 2012).

Line 264: Is there any record on the surface that can support this hypothesis?

Gianni and Navarrete: Yes. We have expanded this part of the discussion by request of reviewer 3. As explained in the text in lines 304-346, there is paleomagnetic evidence of a separated but proximal continental block (North Patagonian massif) in Pre-Permian times (Martinez-Dopico et al., 2020), field-based structural evidence of continental accretion at ca. 285-260 Ma (Review by Ramos et al., 2020), and an igneous record of slab break-off (provided in our study). As noted, a subduction igneous record is missing in this evolution, which would be expected by the consumption of a preexisting oceanic basin before continental accretion. In our interpretation, the latter information supports the hypothesis of Martinez-Dopico et al. (2020) based on paleomagnetic data, where a proximal block separated by a small proto-oceanic basin would have converged, collided, and triggered slab break-off of the available proto-oceanic lithosphere. The latter would have not allowed the existence of arc magmatism, probably because it was not hydrated enough or it was too short to reach typical mantle dehydration depths (100-120 km), but at the same time, the detachment of such slab would explain the existence of slab-break-off magmatism produced by melting of this proto-oceanic slab. By request of reviewer 3 the discussion on Patagonian geology was expanded, which will help to clarify the ideas mentioned above.

Line 284: Please explain further why this scenario is unlikely.

Gianni and Navarrete: We have provided a clearer explanation in this topic in lines 358-362.

Figure 1: It would be better to use a contrast colour to show study area in Figure 1a.

Gianni and Navarrete: The reviewer is right. This was modified as suggested.

Figure 2: 2a has two ChMP polygons on it, are these drawn for two different trench locations for 280 and 260 Ma? If so, it would be better to label or colour them separately to avoid the confusion for the ages they represent.

Gianni and Navarrete: The reviewer is right. This was modified as suggested.

Figure 2e: Why do you think anomaly of the missing slab record in the tomographic images are getting shallower eastward? It reaches 2000 km depth and maybe more. With your calculation this corresponds to 200 Ma.

Gianni and Navarrete: This is because the Mesozoic slab segment that recovered the slab gap in younger subduction stages appeared a bit further west. It can be better pictured if looking at the plan view in Figures 2 a, b, and c.

Fig. S3 is missing in Supplementary file instead Fig. S4 appears twice. I think the Figure caption needs a correction.

Gianni and Navarrete: The reviewer is right. This is now corrected.

Fig. S5: the order of the maps is confusing, it would be better to arrange them in an increasing order or the other way around.

Gianni and Navarrete: We have reordered this figure as suggested.

It has been discussed earlier and in different parts of the text but as a conclusion it would be nice to sum up the ideas in one paragraph. What is the type of the slab failure according to your study? How was it originated? Is there any evidence on the surface, for example the spatial distribution of the magmatics along strike and their age propagation? (by referring to Fig. 4)

Gianni and Navarrete: The reviewer is right. We have relocated and modified a paragraph summing up the main hypothesis and a brief discussion about the ages of slab break-off magmatism (see new lines 391-401).

Reviewer #3 (Remarks to the Author):

Review by Suzanne Mahlburg Kay comments to authors.

This manuscript by Gianni and Navarrete addresses the origin of the silicic Choiyoi magmatic province in South America, which marks a major change in crustal and mantle magma conditions at the Gondwana margin coinciding with the final assembly and initial breakup of the Pangean supercontinent. The topic was addressed some time ago in a series of papers by Kay et al. (1989), Mpodozis and Kay (1990, 1992) and Ramos and Kay (1991), who suggested that the Choiyoi province formed on the South American margin after the major collision of Laurentia with Pangea as subduction ceased along this part of the Gondwana margin with the final terrane collisions leading to slab rupture (break-off) and an associated massive mantle magma upwelling that generated the crustal melting that created the Choiyoi Province (called the ChMP or Choiyoi SLIP in this paper). The Mpodozis and Kay papers argued that important subduction related volcanism did not return to the margin until about the earliest Jurassic marking an important interruption in arc volcanism on or at least on this part of the Gondwanan margin. They supported their hypothesis with the major and trace element and isotopic data available at the time. Subsequently, a series of papers appeared arguing that this was incorrect and that subduction was continuous along the margin throughout this time. Some argued that an eastward sweep in silicic volcanism resulted from steepening of a flat subduction zone. This controversy arose from what Gianni and Navarrete call an “ambiguity” in the geochemical data that comes from analyses plotting in arc fields on discriminant diagrams. Among the problems was the plotting of data with little attention as to the possibility that analyses could reflect extensive melting of older underlying arc crust producing an inherited rather than an active arc signature. In this vein, it is very refreshing to see this paper by Gianni and Navarrete that uses tomographic evidence to define a possible slab gap that is consistent with slab break-off and a major change in geodynamic conditions at the time of Choiyoi magmatism, and that reopens the use of geochemical data in making a case for continuous subduction during this time. Calling attention to the possible distribution of slabs at the time of the Choiyoi from evidence in the lower mantle is an important advance. The appropriateness of the slab configurations chosen and the concept of a “vote map” for selecting a configuration are outside my expertise. A question is whether the tomographic data used in the vote map analyses here remains the most up-to-date as the status of tomographic data is rapidly evolving as new seismic data are incorporated and processing techniques improve (see comment on line 116). The re-raising of the question of the interpretation of the geochemical data from the Choiyoi by Gianni and Navarrete is also a useful exercise and their demonstrating that there is an avenue for an interruption in subduction supported by the geochemical data as suggested by Mpodozis and Kay is a step forward. In this regard, the authors refer to a “discovery of a geochemical signature of slab failure” when they plot a modern compilation of geochemical analyses on a series of x-y chemical plots from Hildebrand et

al. (2018) and Whalen et al. (2019) that have a field for slab breakoff. The basis as to why these multiple and somewhat repetitive diagrams, which are based on long-established geochemical principles, might work could be better explained so the chemistry does not again become a “black box” as in too many previous papers on the Choiyoi. The diagrams are said to provide the evidence that subduction was not continuous and that slab rupture and crustal thickening events are recorded in the geochemical data. Although the interpretation is very reasonable, there is also a danger to these diagrams, as signatures that plot in the slab breakoff field exist in the Andes today. Julian Pearce, one of the early creators of the diagrams often used for the interpretation of the Choiyoi, once said that his greatest regret in publishing these discrimination diagrams was that they were too often misused by those who did what not understand their basis. Outside of the immediate theme of this paper, the manuscript here does not address whether the Choiyoi event has a place in the supercontinent cycle and the relative importance of subduction on the margins of the Pangean supercontinent. Nevertheless, the argument here is in accord with the Mpodozis and Kay (1992) suggestion that subduction was not prominent along the South American margin at the time of Pangea and could be significant in the ongoing discussion of how continuous or discontinuous subduction was along the entire Pangean margin. There is a current fad to write papers like text messages despite the fact that limiting abbreviations makes paper easier to follow for the reader. This paper proliferates this trend using SWPM for South Western Patagonian margin, WPM for Western Patagonian margin, ChMP for Choiyoi magmatic province, SLIP for silicic large ignimbrite province, etc.

Gianni and Navarrete: We are thankful to Prof. Suzanne Mahlburg Kay for this critical and profound revision. We have followed all these suggestions. Detailed responses to each of these comments are provided below in answers to the specific comments keyed to line numbers.

Comments keyed to line number:

Line 5 and 6 in abstract. Consider limiting abbreviations that make papers hard to read. Terms like SWPM for southwestern Patagonian margin, SPM for southern Patagonian margin and ChMP for Choiyoi Magmatic province do not add clarity for readers who just have to remember what

Gianni and Navarrete: The reviewer is right. We have changed SWPM to southwestern Pangea margin, ChMP to Choiyoi Magmatic Province, ET to Equis terrane, SPT to south Patagonian terrane, and NPT to north Patagonian terrane. However, we decided to keep abbreviations regarding the three main geographical regions of the Choiyoi Magmatic Province and the analyzed lower mantle slabs in figures and captions because including all these names in our figures will make them difficult to read.

Line 8. Controversy is largely due to casual misuse and lack of understanding of the chemical data rather than the ambiguity of the data.

Gianni and Navarrete: We decided to remove the sentence claiming that geochemistry was ambiguous. As pointed out by Dr. Kay it is the improper use of it the cause of confusion.

Line 10. Informs is a strange word. How about “the lower mantle slab record beneath the southwestern Pangean margin that provides clues to late Paleozoic-Mesozoic subducting slab configurations, geochronological information and properly interpreted geochemical data.”

Gianni and Navarrete: We have modified these lines (see new lines 7-9).

Line 15. This is a strange statement - “discovery of a geochemical signature of slab failure”. Not a discovery by those who use geochemistry as it should be used rather than just plotting analyses on discriminant diagrams. The whole geochemical data set needs to be considered. A similar case can be made for improper use of seismic tomographic data.

Gianni and Navarrete: The reviewer is right. We have modified this line.

Line 17. Text says findings render the Choiyoi magmatic province the only silicic large igneous province entirely formed by slab failure and the oldest example of a geophysically constrained slab loss event. Although, not geophysically constrained, another could be the Precambrian North America granite-rhyolite province.

Gianni and Navarrete: The reviewer is right. We now refer to this SLIP in new lines 452-454.

Line 27. Kay et al. (1989) used Choiyoi Magmatic Province. The designation ChMP is a recent construct.

Gianni and Navarrete: The reviewer is right. We have now changed this name to Choiyoi Magmatic Province as suggested.

Line 33. The Choiyoi province was called a large silicic magmatic province before 2020, but has not always been designated by the abbreviation SLIP.

Gianni and Navarrete: The reviewer is right. We included a reference to the study of Kay et al. (1989). We also refer to the work of Bastías-Mercado et al. (2020) because it provides the first detailed volumetric estimation for the Choiyoi Magmatic Province and the most recent overview on this topic. Also, we use the term SLIP because it is the most common way to refer to large silicic igneous provinces and hence, the readers will be readily familiarized with this term.

Line 34-39. The referencing in this section seems rather scattered for the points made

Gianni and Navarrete: We think the references are appropriate to the points made. Regarding field evidence of extension-transtension associated with the Choiyoi Magmatic Province: To our knowledge, the only works that have provided concrete structural field data of transtension and/or extension during the development of this magmatic province are those of Giambiagi et al. (2011), (2008) and Rocher et al. (2015). Of course, many articles have used geochemistry to suggest extension-transtension. At this point, we only referred to the work of (del Rey et al., 2019) that interpreted an extensional context based on geochemistry data. Now we included a reference to the study of Mpodozis and Kay (1992) that suggested this setting previously. In the next paragraph, we provide many more references regarding the hypothesis of the overall geodynamic context of the Choiyoi Magmatic Province. Concerning studies providing local evidence of contraction associated with the Choiyoi Magmatic Province: The works we are referring are those of Kleiman et al. (2009) and Chernicoff et al. (2013) that provided field evidence of contraction in the San Rafael region and the North Patagonian Massif. Concerning references of the link between the Choiyoi Magmatic Province and the end-Permian mass extinction, we have added a reference to the recent work of Nelson and Cottle (2019).

Line 46. Has similarities to the Mpodozis and Kay (1990, 1992). See summary figure that shows subduction in the pre-Choiyoi, followed by uplift and collision, and subsequent formation of the Choiyoi province after ~ 250 Ma.

Gianni and Navarrete: The reviewer is right. We have now included those references in this line as suggested.

Line 48. Moreover, this makes perfect sense with the Mpodozis and Kay (1992) model, which was tied to both geologic and geochemical data.

Gianni and Navarrete: The reviewer is right. We have now included in this line a reference to Mpodozis and Kay (1990) that was missing and also includes geological and geochemistry data.

Line 49 - 52. Many of these papers use the geochemistry in an unsophisticated way leading to difficulties in interpretation and lumping analyses on plots without much attention to what the samples are or where they are from.

Gianni and Navarrete: We decided to remove the sentence claiming that geochemistry was ambiguous. As pointed out by Dr. Kay it is the improper use of it the cause of confusion.

Line 55. The isotopic data fit well with the slab break off model of Mpodozis and Kay, who made a point of a temporal isotopic shift based on the isotopic data in their paper and that available in 1992. The isotopic shift was correlated with slab break-off and a switch from a subduction to a post subduction extensional regime. This agrees well with more recent isotopic data used to indicate a shift from a source with a more crustal influence to a larger mantle component in the Choiyoi magmas.

Gianni and Navarrete: We agree with the reviewer. A slab break-off model would also be compatible with the available isotopic data. However, all recent studies have claimed that these data support ongoing subduction and backarc extension. In agreement with the reviewer's comment, our study suggests that isotopic data can also be compatible with the alternative slab break-off model (lines 266-275).

Line 59. Better not to stress the ambiguity of geochemical data - the problem is how the data are used and interpreted – not the actual data.

Gianni and Navarrete: The reviewer is right. We decided to remove the sentence claiming that geochemistry was ambiguous. As pointed out by Dr. Kay it is the improper use of it the cause of confusion.

Line 70-72. And this is how Mpodozis and Kay (1992) interpreted the geochemical data before all became confused on classification diagrams that did not consider the genesis of the magmas. The diagrams here are somewhat repetitious as Y+Nb denote a similar behavior to Ta+Yb, Y+Nb is similar to Ta+Yb, and low Yb contents and high Sr/Y both indicate depletion of heavy REEs, etc. This is the same chemistry that has been used to indicate garnet retention, arc signatures and slab breakoff for years.

Gianni and Navarrete: The reviewer is right. Mpodozis and Kay (1992) interpreted a slab break-off based on geochemical data of the Andean region, so we have referred to this article in several parts of our manuscript. Regarding the repeatability of the diagrams, which consider several

geodynamic settings, our intention is to show that the slab failure signature is observed in most of the diagrams to give more support to our proposal.

Line 74-76. Needed? “Finally, this study illustrates an interdisciplinary approach to solve geological problems testing geodynamic hypotheses in ancient convergent settings back to late Paleozoic time.”

Gianni and Navarrete: This is one of the major contributions of our study as the approach followed to solve the origin of the Choiyoi Magmatic Province could be applied to solve the geodynamic setting of other SLIP, which are intensively debated. A clear example of this is the Chon Aike magmatic province which has been associated with an expanding plume head near the southwestern Gondwana margin (Pankhurst et al., 2000 *J. Petrol*) and more recently interpreted as a wide arc region (Bastías et al., 2021, *Lithos*).

Line 92. The term tomotectonics seems to have come into the literature in ~2017 and has caught on since. A minor point, but this term is not used in the references cited here.

Gianni and Navarrete: Yes, the reviewer is right. This type of methodology was formally referred to as tomotectonic analysis by Sigloch and Mialiniuk (2018). As this term has been used since then, we decided to use this term in our study.

Lines 116 and 128. Please briefly explain what you mean by a vote map and vote map stacking, as many readers will not be familiar with this methodology. Maybe you could move the discussion on vote maps from the methods section on lines 365 to 172 to this point in the paper.

Gianni and Navarrete: The reviewer is right. We have introduced additional lines addressing this point in new lines 124-128 as suggested.

There are a series of recent papers on tomographic models looking at slabs at depths of 200 to 1200 km using P-wave tomography [Portner et al. 2019, *JGR* & Mohammadzaheriet et al 2020, *JGR*] and S-wave tomography (Rodriquez et al. 2021, *Geophys. J. Int*]). How well these tomographic do models match and mesh with the P and S wave tomographic models at greater depths featured here? Particularly, could any these of these older lower mantle models be affected by the more recent increase in seismic tomographic data available for South America? Just a question.

Gianni and Navarrete: We chose the UU-P07 (Amaru, 2007) for our reconstructions because is one of the most powerful models to detect lower mantle slabs. It is likely the most widely used global

seismic tomography model for tomotectonic analyzes (van der Voo et al., 1999, uses the first version of this model, Science; 2015, Geological Soc of Lond.; van der Meer et al., 2010, 2012, Nat. Geosc. 2018, Tectonophysics; van Hinsbergen et al., 2019, Tectonophysics; 2020, GRL; among many others) (this is was indicated in new lines 93-95). Nevertheless, as explained in our article, our analysis is not limited to one seismic tomography model. We carried out a vote map analysis that includes 26 of the most recent global seismic tomography models (P and S-wave) with most of them differing in data selection and parametrization, and regularization of the inversion (GyPSuM-S, Simmons et al., 2010; DETOX P2 and P3, Hosseini et al., 2020; HMSL-P06 and S06, Houser et al., 2008; PRI-P05 and -S05, Montelli et al, 2006; SPani-P and -S, Tesoniero et al., 2015; GAP-P4, Owayashi et al., 2013; LLNL_G3Dv3, Simmons et al., 2012; Hosseini2016, Hosseini, 2016; SEISGLOB1, Durand et al., 2016; MITP08, Li et al., 2008; UU-P07, Amaru, 2007; TX2019Slab-P and S, Lu et al., 2019; S362ANI+M, Moulick et al., 2014; S20RTS, Ritsema et al., 1999; S40RTS, Ritsema et al., 2011; SAVANI 32, Auer et al., 2014; SAW642ANb, Panning et al., 2010; SEMUCB-WM1, French et al., 2014; SEMum, Lekic et al., 2014; TX20111, Grand et al., 2002; TX2015, Lu et al., 2016; see model details in 33 Table S1). Please note that tomotectonic studies often compare a much smaller number of seismic tomography models. This analysis shows that the slab gap detected in our reconstructions with the UU-P07 model is also common to most of the 26 seismic tomography models making our choice of the UU-P07 model appropriate and our interpretations about the mantle structure robust. Concerning the seismic tomography models used in recent studies analyzing the South American mantle structure above 1200 km. Chen et al. (2019), based their main analysis on the MIT08 seismic tomography model (Li et al., 2008), which is included in our vote map analysis. Mohammadzaker et al. (2020) is essentially based on the DETOX-1 model. As we are interested in depths below 1200 km, in our vote map analysis we included the recent DETOX-P2 and P3 models, which are more suitable to analyze the deeper lower mantle structure (Hosseini et al., 2020). Finally, as explained in Portner et al. (2019) and Rodriguez et al. (2021) the SAM5_P_2019 model, and the updated version SAM5 S 2020, are not suitable for mantle depths below 1500 km, and hence we do not include them in our vote map analysis, which explores deeper mantle regions.

Line 120. Data base in file S1 used here is from references 102-123 and is a diverse data set from a wide area. You could add a few more comments in the text as to what is included in this data set and why. You do not reference the much smaller 'less complete' data set included in and compiled by Kay and Mpodozis (1992) from which they arrive at generally similar conclusions.

Gianni and Navarrete: Data from Kay and Mpodozis (1992) are used for the Late Choiyoi magmatism in our analyses (see supplementary dataset S1). However, we could not include all the data from the latter study because several units used by Kay and Mpodozis (1992) do not have precise dates, so the emplacement age is not well known. For example, Kay and Mpodozis (1992) used data from the unit called "Guanta" and although it has a U-Pb age of 285 Ma (Pankurst et al., 1996), there are also older ages (~ 300 Ma) and more modern (~ 260 Ma) geochronological data

for the same unit. Likewise, available ages for the "Montosa" have a high uncertainty (~290-240 Ma. see Mpodozis and Kay, 1992 for a review). As our intention was to use more precisely constrained units, these data were not used in our analyses.

Line 131. Tomographic data are from references numbered 39 and 40 - Van der Meer et al. (2012, 2018). Maybe you could cite the author's names as the Nature referencing system puts the burden on the reader to search for the reference by number.

Gianni and Navarrete: This was modified as suggested.

Line 135. Same comment as 131. Say that data is from Spikings et al. (2015) and Chew et al. (2016).

Gianni and Navarrete: This was modified as suggested.

Line 142. Text says reconstructions in Fig.1a, b with tomographic depths. Should this be Figure 2?

Gianni and Navarrete: The reviewer is right. We have corrected this as suggested.

Line 159. Note partial melts of preexisting arc rocks can lead to arc signatures, but this does not necessarily signify these magmas formed in an active arc environment. This problem with plotting silicic rocks on discrimination diagrams has long plagued the interpretation of the Choiyoi. Why show plots from both Hildebrand et al. and Whalen and Hildebrand when you do not discuss why these specific elements are plotted. The first set is largely based on the HFSE and the second on the slope of the REE patterns. Slab failure leads to depleting the heavy REE as indicated by low Yb and Y (depleted HREE), high Gd/Yb ratio and elevated Sr contents - all of which indicate a garnet signature with a breakdown of plagioclase. Mpodozis and Kay (1992) pointed out these signatures in their Cochiguas and El Vulcan samples. Note also that the decrease in $^{87}\text{Sr}/^{86}\text{Sr}$ in Fig 3 of Kay and Mpodozis indicates a less enriched isotopic source after the Cochiguas and El Vulcan Units.

Gianni and Navarrete: The reviewer is right. We now explain in more details in the method section the principles followed by Hildebrand et al. (2018) and Whalen and Hildebrand (2019) to build these tectonomagmatic discrimination diagrams. The latter will provide the reader with a clearer idea of the approach followed by those authors and how these diagrams should be used and interpreted in future studies.

We are aware that signatures that plot in the slab breakoff field exist in the Andes today. However, as we argue in new lines 276-285, in thick orogenic crusts such as in the Central Andes and the Tibetan plateau, high and low values in La/Yb, Sr/Y, Gd/Yb, and Sm/Yb ratios are found at any time (see Mamani et al., 2009 Geol. Soc. Am. Bull.; De Paolo et al., 2019; Gondwana Res.) because fractional crystallization and crustal assimilation can also occur at shallow crustal depths (a discussion on this topic can be found in Worner et al., 2018, Elements). The latter would result in diagrams with values plotting in both arc and slab-failure fields, which contrasts with the results of the Choiyoi magmatic province.

Line 163-165. La/Yb versus Yb diagrams – back to adakites and the name that means little more than an undefined high Sr content and a low HREE content indicated by Yb or Y that together imply garnet retention and feldspar breakdown. As a comment, most of these adakites look little like the Adak lava in the Aleutians from which Drummond and Defant took the name. A question - what at the nature of the samples that plot with La/Yb ratios near 100 on Figure S8

Gianni and Navarrete: The reviewer is right, the high La/Yb ratios indicate the garnet retention and feldspar breakdown, which could be obtained from the partial melting of an eclogitized subducting slab, or by the partial melting of a lower eclogitized crust; as well as by the partial melting of a delaminated eclogitized crustal portion or by high-pressure fractional crystallization (see Zhang et al., 2019 for a review). In the Choiyoi Magmatic Province case, there is no concrete evidence to indicate that the magmatic emplacement and extrusion occurred in a thickened crust or was affected by a delamination process. This is particularly evident for the Andean region and the intraplate magmatic belt where an extensional regime has been described during the Choiyoi Province by field studies (Giambiagi et al., 2008, Martínez and Giambiagi, 2010; Geosphere; Rocher et al., 2015; Rev. Geol. Mex.; Kleiman et al., 2009). In the intraplate magmatic belt, Llambías et al. (2003) and Chernicoff et al. (2018), highlighted an insignificant influence of the San Rafael compressional phase, and a thin to normal crust was then inferred by those authors at the time of Choiyoi emplacement. The crust in the intracratonic region is below 40 km (Tassara et al. 2006; JGR) and there is no geological evidence so far for the existence of a thicker crust in the past. Similarly, for the North Patagonian Massif region, an orogen collapse has been suggested for the Late Permian-Early Triassic stage (Ramos, 2008 and references therein, Fanning et al., 2011; Luppo et al., 2016). We have now introduced a brief comment addressing the above-mentioned points in new lines 276-285.

The samples with La/Yb ratios near 100 are acidic rocks (rhyolites) with ~69-70% SiO₂.

Fig. 3 - Line 803. Note typo – should be 261-246 Ma, not 61-246 Ma.

Gianni and Navarrete: The reviewer is right. We have corrected this as indicated.

Line 176. Eastward broadening of Choiyoi province – up to 500 km from current trench, much is actually in today's forearc. What does this mean considering strike slip and the possible effects of forearc subduction erosion?

Gianni and Navarrete: The reviewer is right. We now consider subduction erosion in our spatio-temporal diagram as suggested by the reviewer. For this correction, we consider a long-term (70 Ma) average subduction erosion rate of 1 cm/yr calculated by Kukowski et al. (2006) for the Andes. Of course, it is a simplification as subduction erosion has been more effective and punctuated at certain times with variable influence at different latitudes (e.g. Kay et al., 2005). However, the latter correction should provide a more precise spatio-temporal analysis than not considering subduction erosion at all. As the magnitudes of strike-slip motion in the forearc remain largely unknown, we can't consider this process in our analysis, and we indicate so in the text (new lines 145-150). Also, we did not consider Andean shortening, which varies along strike the orogen, and although better quantified than forearc strike-slip motion, it is still unknown in some areas. Hence, plotted arc-to trench distances represent a minimum value of magmatic migration. For our purposes, this simple analysis is enough to demonstrate an important inland magmatic migration, which is our main point to discard the slab steepening hypothesis that would produce a trenchward magmatic migration. The limitations in our analysis are now flagged to the reader in lines 148-150.

Line 185. How clear is this arc signature and how extensive is it in late Triassic and Early Jurassic igneous rocks? Silicic rocks can have significant inherited crustal components whose presence do not confirm they erupted in an active arc. I have not read the papers recently, but I don't recall overwhelming cases for extensive arc rocks in Choiyoi magmas proper in papers denoted by references 23 (Oliveros et al.), 29 (Lopes de Luchi et al.) or 59 (Vasquez et al.). Figure S10 gives no indication of silica content or evolutionary stage of the plotted samples and case needs to be made based on widespread mafic samples.

Gianni and Navarrete: We understand the reviewer's point of view, and we share the opinion on the difficulty of determining arc magmatism. We based our conclusions on the following. The geochemical diagrams of Hildebrand et al. (2018) and Whalen and Hildebrand (2019) show an increase in the arc geochemical signature compared to Permian-Early Triassic magmatism (Fig. S10). The Sr/Y, Nb/Y, La/Yb, Gd/Yb, Ta/Yb, and Sm/Yb ratios are lower than the ChMP, suggesting a change from a garnet-bearing plagioclase-free source to a garnet-free plagioclase-bearing melts source, which is typical for the arc magmatism. We understand that this can also be an inheritance if crustal melting involved a previous arc basement because Hildebrand et al. (2018) and Whalen and Hildebrand (2019)'s diagrams use rocks with silica contents between 55 and 70%. However, isotopic studies revealed values of ϵ_{Hf} between 0 and 7 and mantle-like values of $\delta^{18}\text{O}$ (3,7-6,5), which have been interpreted as indicating the presence of a more juvenile and depleted magma source after ~220 Ma that along the arc-signatures in these rocks was collectively interpreted as associated with a mantle wedge source in a subduction setting (see synthesis on del Rey et al.,

2016). In support of this view, we found that the geochemical shift to arc signatures correlates with a recovery of the active margin in the tomotectonic analysis. Our analysis reveals a slab beneath the reconstructed South American convergent margin at these times (Fig. 2c). We consider this an important point because this analysis provides an independent constraint on the subduction setting at this time. Therefore, we are inclined to think that subduction is activated sometime between Late Triassic and Early Jurassic. The latter is indicated in new lines 388-390.

Line 207. This is pretty similar to the slab breakoff hypothesis in Mpodozis and Kay (1992). The difference is that you restart the arc sooner – but what is the real evidence in the geochemistry as to when normal arc volcanism reappear

Gianni and Navarrete: This is answered in our comment above.

Line 210. Please review the evidence that Chernicoff et al. offered for removing the Intracratonic magmatic belt from the Choiyoi magmatic province. How is Figure 3 relevant to this argument?

Gianni and Navarrete: The reviewer is right. We have modified these sentences accordingly.

Near line 215. The steepening of a shallow subduction zone was always a difficult model to visualize particularly when an analogy was made with the modern Puna and ignimbrite flare-up, which is a very different story from the Choiyoi.

Gianni and Navarrete: We agree 100%. With our spatio-temporal analysis in Fig. 4 indicating an inland magmatic migration associated with the Choiyoi magmatic Province, we provided additional arguments for discarding the slab steepening hypothesis, which is the best candidate for trenchward arc migration. Even without taking into account our seismic and geochemistry analyses, this simple but key argument is enough to discard the slab steepening hypothesis in the Andean region of the Choiyoi Province.

Line 217. Again, Mpodozis and Kay (1992) argued against this scenario, called for slab break off and a break in subduction with the evolution of the Choiyoi Province being closely linked with events related to Pangea. Do you think there is any connection with the Pangean supercontinent or is the Choiyoi province just an unrelated episode of slab break off that happens to occur on the Pangean margin?

Gianni and Navarrete: We understand the point of view of the reviewer. However, we do not think that the slab break-off process has something to do with the assembly of Pangea. As seen in our reconstructions the slab gap formed in a restricted area coinciding today with the Central Andes and Permian slabs are reconstructed south and north of this area. If Pangea indeed terminated subduction we would not expect the latter observations. However, we now include the

insightful observation of Kay et al. (1989) that Pangea thermal insulation would have provided an additional source of heat for extensive melting associated with the Choiyoi magmatic Province (see new lines 293-298). Most recently, numerical and geochemical studies have found support for this process indicating upper mantle excess temperature up to 150°C (Coltice et al., 2008, EPSL; Brandt et al., 2013 Nat geosc.).

Line 227-243. In other words melting of eclogite. How about a component from partial melting of the lower crust by mantle-derived magmas? Again, this is more or less what Mpodozis and Kay argued – they did not mention high-K mafic magmas. The huge improvement over their model is your perspective on the slab gap based on evidence from the lower mantle.

Gianni and Navarrete: The reviewer is right; we added melting of eclogite in this sentence. We mention a component of partial melting of the lower crust in new lines 286-288.

Line 245-275. Discussion of Southern and Northern Patagonian terrane accretion. This section is hard to follow for those not familiar with the latest on this ever changing and seemingly never ending debate. Need more context for the reader to follow this.

Gianni and Navarrete: The reviewer is right. We have included several sentences expanding this topic as suggested by the reviewer (see new lines 304-322).

Line 258. Autochthonous in what time frame?

Gianni and Navarrete: In Early-mid Permian. We now indicated this in this part of the discussion. In the hypothesis of Pankhurst et al. (2006; 2016) and Fanning et al., (2011), the North Patagonian massif is autochthonous in Permian times and the Permian arc suggested by Ramos in his many studies is instead interpreted as slab break-off magmatism.

Line 259. Is all of this discussion based on analyses falling in a field labeled slab failure? How do you explain the paleomag data? The discussion of the North Patagonian Terrane needs more context.

Gianni and Navarrete: All this is now expanded in new lines 304-322. Also, in this new paragraph, we explain that new paleomagnetic data supports hypothesis of a short-traveled para-autochthonous origin for the North Patagonian terrane indicating a location of Patagonia between 450 and 250 Ma close to its present location.

Line 275. There has been too much emphasis on the details of the Equis terrane as a means to detract from the Kay and Mpodozis (1992) proposal for the termination of subduction with slab break-off. The slab failure event was never said to be restricted to the region of the Equis terrane. A way was needed to stop subduction and a possibility was to do this with terrane collisions in the final amalgamation of Pangea. The Equis terrane was proposed to end subduction along part of the south central Andes (and explain the San Rafael orogeny) and was fashioned on a proposal in Australia. Although speculative, all of the evidence for the Equis and similar terranes could be absent due to subsequent forearc subduction erosion. How is lack of concrete evidence for a hypothetical X terrane (equis in Spanish) an argument against the slab break-off model of Mpodozis and Kay? What is the alternative evidence for subduction of an ocean ridge and how does this work with the San Rafael event? Seems more tenuous than a hypothetical terrane removed by forearc subduction erosion. The whole issue leads to the larger question of whether or not there was subduction along the margin of Pangea existed and the influence of supercontinents. The seismic evidence featured here could be key in examining this issue.

Gianni and Navarrete: The reviewer is right. We now argued in lines 350-352 that the lack of basement outcrops could be due to forearc subduction erosion as suggested early by Mpodozis and Kay (1992). Regarding the inferences on the past existence of aseismic ridge-trench interactions, in the last years, several studies have indicated the presence of blue and greenschist with mafic protoliths of OIB and E-MORD affinities accreted between 300 and 280 Ma to the Chilean Coastal Cordillera. In Fig. 1b we now show the location along the Andean Margin where these rocks with oceanic affinity have been documented.

see a review in: J. Díaz-Alvarado, G. Galaz, V. Oliveros, C. Creixell, M. Calderón, In *Andean Tectonics*, 576 B. Horton, A. Folguera, Eds. (Elsevier, 2019), pp. 509-530.

Lines 291-298. Accretionary fragments associated with the end of subduction along the margin of the Pangean super continent makes some sense and could include small terranes with the speculative Equis terrane being one of these fragments. Need to take into account that these are final events in the formation of Pangea and the beginning of the incipient break-up and that this is a special time in Earth history.

Gianni and Navarrete: The reviewer is right. We now indicate that slab break-off triggered by the interaction of seamounts support the notion of a series of Equis terranes formed by several oceanic pieces accreted to the southwestern margin of Pangea instead of the docking of a single continental fragment.

Line 304. This discussion more or less supports the concept of a series of “Equis” terranes.

Gianni and Navarrete: The reviewer is right. This hypothesis is now included in lines 380-382.

Line 319-321. There was also early discussion of this in comparing the Southern America Choiyoi-Chon Aike provinces with the North American granite-rhyolite province in Kay et al. (1989) and Mpodozis and Kay (1990, 1992). The point was that massive continental rhyolite provinces are found in relatively young crust and are likely critical in producing stable crust– they are not common in older continental crust.

Gianni and Navarrete: We have now added a reference to Kay et al. (1989) in this sentence as suggested.

Line 328. If you read through the lines, slab failure is also suggested from Peru to Australia in the Mpodozis and Kay (1990, 1992) papers. What specifically are you referring to when you mention enormous amount of andesitic rocks in the lower portion of the Choiyoi magmatic province. Is this really pre-Choiyoi Paleozoic subduction and a confusion on how the Choiyoi province is defined?

Gianni and Navarrete: This is a result of the recent volumetric and compositional estimation of the Choiyoi Magmatic Province by Bastías-Merado et al. (2020). They found that volcanic products of Choiyoi Magmatic Province are composed of 51% rhyolite, 26% dacite, 22% andesite and 1% basalts, but the early stages of the volcanism were dominated by andesitic products. The plutonic rocks make up the 25% of the total area of the Choiyoi province and they are dominated by granites (~65%) and granodiorites (~30%), although the units representing the early stages of the Choiyoi province are composed up to 60% of granodiorites, tonalites and diorites.

Line 380. Why is this discussion put at this point in the paper? It would be better if it was where the topic is highlighted in the paper.

Gianni and Navarrete: The reviewer is right. We have moved this part to the main text as suggested.

Line 741. Reference 1122. Note Rapela is repeated.

Gianni and Navarrete: This was corrected.

Figures.

Fig 1. Could write out names for AR, IMP and ET (sounds like the extra-terrestrial in the movie ET) – there is room on the figure. The Equis terrane proposed by Mpodozis and Kay actually went further south than shown as it was largely based on a transect at 29° to 32°S.

Gianni and Navarrete: This was corrected. As we now followed the view of Equis terranes as a family of small oceanic fragments present in the Carboniferous-Earliest Permian accretionary prisms (as suggested by Prof. Kay), we decided to remove the suture zone tentatively suggested in our previous map.

Figure 4. Careful of what plotting ages of silicic rocks at the current distance to the trench actually means. What does this really mean as there has been a lot of strike-slip motion and possible forearc subduction erosion along the margin since the Choiyoi. As the modern volcanic line is ~ 300 km from the trench, many of the localities are in the modern forearc and only a small portion in the modern back-arc.

Gianni and Navarrete: The reviewer is right. We now consider subduction erosion in our spatio-temporal diagram as suggested by the reviewer. For this correction, we consider a long-term (70 Ma) average subduction erosion rate of 1 cm/yr calculated by Kukowski et al. (2006) for the Andes. Of course, it is a simplification as subduction erosion has been more effective and punctuated at certain times with variable influence at different latitudes (e.g. Kay et al., 2005). However, the latter correction should provide a more precise spatio-temporal analysis than not considering subduction erosion at all. As the magnitudes of strike-slip motion in the forearc remain largely unknown, we can't consider this process in our analysis, and we indicate so in the text (new lines 145-150). Also, we did not consider Andean shortening, which varies along strike the orogen, and although better quantified than forearc strike-slip motion, it is still unknown in some areas. Hence, plotted arc-to trench distances represent a minimum value of magmatic migration. For our purposes, this simple analysis is enough to demonstrate an important inland magmatic migration, which is our main point to discard the slab steepening hypothesis that would produce a trenchward magmatic migration. The limitations in our analysis are now flagged to the reader in lines 148-150.

We are grateful with the three reviewers for providing valuable feedbacks to our manuscript.

Dr. Guido Gianni and Dr. César Navarrete

REVIEWER COMMENTS

Reviewer #1 (Remarks to the Author):

The authors have made a good effort to address the comments of all reviewers, and the revised manuscript is improved as a result.

My main reservation of the paper remains - I am skeptical that the analysis presented is particularly groundbreaking (relative to what I'd expect for a Nature Communications article). For the previous version, I made the comment that the content was generally more suitable for a domain-specific journal. To change my opinion on this the authors would need to bring some new data or analysis, for example a different set of observations that substantiate one of the key points of their work; a more comprehensive analysis of the global SLIP record or slab break-off events leading to new figures that advance our understanding of this process; or, some kind of novel uncertainty analysis to illustrate the robustness of their tomographic interpretation (ie excluding other possible interpretations, although I understand this is hard, and that really we have to accept that relating deep seismic anomalies to subduction ~ 250 Myr will always be rather speculative). In the revision, the new content is essentially a more detailed discussion of the same results presented previously. Consequently I now understand their idea a little more, but don't see new analysis or figures that demonstrate a wider significance to the study, so I don't have much reason to change my previous opinion. However, it is nonetheless the case that the study has been carried out in a logical manner within the limits of the observations and methods available, and I accept that the concept of 'impactfulness' is rather subjective.

Specific comments:

Regarding slab break-off events and SLIPs, the revised version now more thoroughly evaluates other SLIPs in the geological record, but I still wonder about other slab-break off events that must be well-known in recent times (e.g. Cenozoic). If slab break off is an important driver of SLIP production, why is it that apparently many slab break-off events aren't associated with distinct SLIPs? This is another example of the kind of issue that could make the paper more attractive to a broad issue, if the authors can do more analysis to substantiate this rather than just further discussing the current, regional analysis.

A possible consequence of slab break-off (as opposed to other subduction histories) that is not yet considered is the vertical motions of the Earth's surface due to associated mantle flow (ie, dynamic topography). See for example many papers by Gurnis et al, and many others. It may be that the time-dependence of the level of marine inundation within the region of Gondwana adjacent to the study area could tie in with the interpretation of subduction, slab break-off, and re-establishment of subduction.

Regarding the color scale for the vote map (figure 2d): I commented previously that the colormap should be changed, and the authors have adopted a different colormap, avoiding the use of a diverging colorscale. However, although the revision uses a new color map, the alternative version is still diverging and so suffers from exactly the same problem as the previous one. To be specific, an example of an appropriate color map would be one that uses a bright color for the highest value (26?) and gradually becomes less bright continuously towards zero (note that the difference between 0 votes and 1 vote is no more or less significant than the difference between 1 and 2 votes, yet the colormap used here emphasises the region with few (but not zero) votes in bright red, and the boundary between 1 and 0, despite the fact that this boundary has no special meaning).

Reviewer #2 (Remarks to the Author):

Dear editor,

Hereby I send you my review of Guido M. Gianni and César R. Navarrete's paper 'A catastrophic slab loss in Pangea preserved in the lower mantle structure and the igneous record', resubmitted for publication in Nature Communications.

The authors took the previous comments seriously and corrected the necessary parts as requested. They have taken most of the corrections of reviewer #1 and 3, also they tried to reply all their questions in detail. The answers that are given for the comments are satisfactory. Figures are clearer and better in general terms.

This paper shows us that the deeper tomographic anomalies can still be linked to their surface manifestations, and it brings another constraint to the plate reconstructions for the early stages of Phanerozoic. With few minor corrections/additions I recommend this paper to be published.

-In this paper, it is still not clear where the anomaly of Choiyoi magmatic province, slab gap, is located in present-day plate configuration. It can be deduced from Fig. S2 but as I mentioned earlier, a small text about the displacement between the present-day location of the Choiyoi Magmatic Province and the slab gap can also give insight to the reader who is interested in the relative motion between the lithosphere and the average mantle.

-Misspellings in Fig. 5, "North" Patagonia Terrane

-Line 320: How close is it?

-The use of early/late should be corrected in the main and supplementary text according to the following rule (Line 144, 389 etc.). Time-rock (chronostratigraphic) units (lower/upper) should be used for attribution of age to rocks, formations, biostratigraphic zones, unconformities, and seismic reflectors and, consequently, on well logs, columnar and seismic sections, cross-sections, and stratigraphic correlation diagrams. Time-rock units have a base, a top, and a thickness.

Time (geochronological) units (early/late) should be used for describing the time of occurrence of historical events such as periods of erosion, transgression, folding, faulting, faunal extinction, and oil generation and migration and, consequently, timing of events shown on graphical representations of geological history.

Reviewer #3 (Remarks to the Author):

Overall the authors have done a great job in replying to the reviewer's comments and modifying the paper, which is now much stronger and worthy of publishing. The revised manuscript does a very nice job of relating the issues, explaining the methodology and is now an important contribution not only on the origin of the Choiyoi province, but also on combining seismic tomography, geochemical data and geologic observations. The resulting manuscript is a step forward on mechanisms that can create a major silicic igneous province (SLIP), supports an important break in subduction along a section of the margin of the Gondwana supercontinent and opens the door for future discussions.

A few minor comments on the age range of the Choiyoi province and figures are below. The paper might also benefit from being edited one more time for English expression as there are several places where it could be improved. Note also that the number notations to the references for the geochemical data in the supplementary data need to be updated as the numbers are from the initially submitted paper, not the revised version.

A few specific comments keyed to line number.

Line 6. In the abstract, the Choiyoi province is given a specific age between 286 and 247 Ma. As the term "Choiyoi province" has been used loosely in terms of a time range, the range of ages is important as the limits have implications as to how much of the early plutonic and later silicic magmatism is included. In contrast to the abstract, line 30 of the paper says ca. 286 and 247 Ma and the discussion section later in the paper calls for diachronies in events. A short discussion on the time frame could add clarity.

Line 111. Just a thought, but what is the relation of this to the old polar wandering curves discussed in relation to the Choiyoi province by Kay et al. (1989) and Ramos and Kay (1991)?

Line 263. Is this 900 km from the current trench or a trench at the time of the Choiyoi event? Please clarify.

Figure 1. Indicate in caption that when first mentioned that GI is Georgia Islands and SF is Sao Francisco or better yet write out the names on the figure.

Figure 3. Outline for outcrop distance from current trench is not clearly distinct from that from that from the paleo-trench making figure somewhat confusing.

In figure 4, it might be more logical to have the III-IV cross-section, which is further south and more complex, as the third part of the figure and the I-II section, which is further north in the middle of the figure. Exchange labels III and IV with I and II.

RESPONSE TO REVIEWER COMMENTS: In general, Gianni and Navarrete did an impressive job in responding to the reviewer's comments.

I did find one strange comment in reference to my own papers in the author's response to the comment of Reviewer 1 on the Nelson and Cottle (2019) paper. In section 1 of their reply, I was surprised to see Kay et al. (1989) and Mpodozis and Kay (1992) included among the papers calling for an uninterrupted subduction zone setting for the Choiyoi province. Initially, it left me wondering what they meant by an "uninterrupted" subduction zone setting as our papers clearly called for a break in the subduction process. I concluded it was a glitch in the response as it conflicts with what is said in the text.

Response to reviewer's comments on the manuscript entitled "**A catastrophic slab loss event in Pangea preserved in the lower mantle structure and the igneous record**" by Guido M. Gianni and César R. Navarrete for consideration in *Nature Communications*

REVIEWER COMMENTS

Reviewer #1 (Remarks to the Author):

The authors have made a good effort to address the comments of all reviewers, and the revised manuscript is improved as a result.

My main reservation of the paper remains - I am skeptical that the analysis presented is particularly groundbreaking (relative to what I'd expect for a Nature Communications article). For the previous version, I made the comment that the content was generally more suitable for a domain-specific journal. To change my opinion on this the authors would need to bring some new data or analysis, for example a different set of observations that substantiate one of the key points of their work; a more comprehensive analysis of the global SLIP record or slab break-off events leading to new figures that advance our understanding of this process; or, some kind of novel uncertainty analysis to illustrate the robustness of their tomographic interpretation (ie excluding other possible interpretations, although I understand this is hard, and that really we have to accept that relating deep seismic anomalies to subduction ~250 Myr will always be rather speculative). In the revision, the new content is essentially a more detailed discussion of the same results presented previously. Consequently I now understand their idea a little more, but don't see new analysis or figures that demonstrate a wider significance to the study, so I don't have much reason to change my previous opinion. However, it is nonetheless the case that the study has been carried out in a logical manner within the limits of the observations and methods available, and I accept that the concept of 'impactfulness' is rather subjective.

Gianni and Navarrete: We appreciate the comments by reviewer 1. As required by the reviewer, we now present two additional analyses that substantiate the two key points of our work and demonstrate a wider significance to the study.

Following the reviewer suggestions in the main comment and the specific comments below, we included an additional paleogeographic analysis of the southwestern Pangea margin. For this, we compiled a series of paleogeographic maps that show major paleogeographic modifications of the southwestern Pangea margin that are compatible with a large-scale slab break-off event in the Permian and subduction reactivation in the Late Triassic- Jurassic (see new lines 428-458 and new Fig. 6). We provided a detailed description of this point and references to the studies used to build the paleogeographic maps in the specific comments below.

Also, as suggested by reviewer 1, we added a tomotectonic analysis for the three pre-Cenozoic SLIPs mentioned in the discussion section (Jurassic Chon Aike, Lower Cretaceous Whitsunday, and Upper Okhotsk-Chukotka SLIPs). This is intended to illustrate how linking plate reconstructions and the mantle structure can provide independent information to geochemistry to unravel the debated geodynamic setting of ancient SLIPs (see new lines 478-489 and figures S10 and 7).

We are thankful to reviewer 1 for encouraging us to push further our analyses and the implications of our study.

Specific comments:

Reviewer #1: Regarding slab break-off events and SLIPs, the revised version now more thoroughly evaluates other SLIPs in the geological record, but I still wonder about other slab-break off events that must be well-known in recent times (e.g. Cenozoic). If slab break off is an important driver of SLIP production, why is it that apparently many slab break-off events aren't associated with distinct SLIPs? This is another example of the kind of issue that could make the paper more attractive to a broad issue, if the authors can do more analysis to substantiate this rather than just further discussing the current, regional analysis.

Gianni and Navarrete: This is a good point. We thank reviewer 1 for the opportunity for further clarification in our manuscript. We did not intend to suggest that all slab break-off processes drive SLIPs, but rather we indicated that large occurrences of silicic magmatism are favored by fortuitous events where multiple slab break-off triggers act in concert such as the multiple oceanic and continental terrane accretions linked to the Choiyoi province. Furthermore, we make an additional emphasis on the role of supercontinental thermal insulation for the occurrence of slab break-off SLIPs. As previously suggested by Kay et al. (1989) for the Choiyoi SLIP, an additional heat source enhancing partial melting in this region could have been the upper mantle warming in excess of 150°C produced by supercontinental thermal insulation (a process now supported by numerical modeling and geochemical data). This may enhance slab break-off magmatism and consequently, it is possible that slab break-off SLIPs were more common during supercontinent assemblies and initial break-up stages. If we accept that plate tectonics was an ongoing process in Archean-Paleoproterozoic times, weaker slabs along with thicker oceanic crusts and high mantle potential temperatures (150-250°C) at this time could have favored slab break-off flare-ups (see discussion in Hildebrand et al., 2018). This point is now discussed in lines 498-509.

Also, as described in the manuscript, we do not intend to extend this process to the origin of all SLIPs. As already suggested, SLIPs may form in different geodynamic settings (Bryan, 2007) (see lines 465-471). We emphasize that the analysis of the mantle structure coupled to state-of-the-art plate kinematic reconstructions is a powerful tool to shed light on the geodynamic setting of ancient SLIPs, which is often masked by the highly contaminated rocks in the latter. Following the suggestion of reviewer 1, we did additional studies including the examples of Pre-Cenozoic SLIPs mentioned in our study, and illustrate how this analysis can effectively provide insights into the debated origin of SLIPs (see new lines 478-489 and figures S10 and 7). To avoid including another large figure, we decided to separate the analysis into two figures, one with the original

tomotectonic analysis located in the supplementary material in Fig. S10 and the other with the interpretation presented in the new figure 7.

Reviewer #1: A possible consequence of slab break-off (as opposed to other subduction histories) that is not yet considered is the vertical motions of the Earth's surface due to associated mantle flow (ie, dynamic topography). See for example many papers by Gurnis et al, and many others. It may be that the time-dependence of the level of marine inundation within the region of Gondwana adjacent to the study area could tie in with the interpretation of subduction, slab break-off, and re-establishment of subduction.

Gianni and Navarrete: The reviewer is right. We appreciate this insightful suggestion. We had not considered the possible paleogeographic modifications of southwestern Pangea in the Permian-Triassic and the potential relation to changes in the dynamics of the convergent margin proposed in our study.

In the new manuscript, we compiled paleogeographic data shown on top of plate kinematic reconstructions of Pangea from Matthews et al. (2016). Paleogeographic data has greater detail for basins in the study area. For the study area, we utilized paleogeographic data from Limarino and Spalletti (2006) and Limarino et al. (2014). We used paleogeographic data from Ford and Golonka (2003) and Torsvik and Cooker (2013) for the regions outside the study area.

As conceived by reviewer 1, a series of paleogeographic maps in new figure 6 shows a striking relationship between the margin inundation history and the suggested evolution from slab break-off to subduction reactivation. The slab loss stage associated with the Choiyoi Magmatic province coincides with a long recognized forced regression and a marine disconnection between Panthalassa Ocean and backarc basins along the Andean region (Limarino & Spalletti, 2006). Furthermore, it coincides with the progressive development of a Late Permian positive relief that divided the western Pangea margin in a western template region under the marine influence and an inland eastern area where semiarid and arid conditions prevailed (Limarino et al., 2014) (Fig. 6a).

To date, how this positive topography was formed along the Andean region is still poorly understood. Furthermore, it is difficult to explain why backarc basins were disconnected from the Panthalassa Ocean during a period of relatively high global sea level (Guillaume et al., 2016) (Fig. 6b). Our findings provide clues into these issues. Conceptual and numerical studies indicate that slab-break-off can drive rapid surface uplift, which may last between 10 and 20 Myr (Davies & von Blanckenburg, 1995; Duretz et al. 2011). We suggest that the topographic uplift produced by the massive slab break-off associated with the Choiyoi magmatic province provides a straightforward explanation for the disconnection of backarc basins from the Panthalassa Ocean and the Permian topographic changes (Fig. 6a). Also, as conceived by reviewer 1, the subduction reactivation stage in the Late Triassic-Early Jurassic coincides with the recovery of marine inundation of the southwestern Pangea margin (see Vicente, 2005 for a synthesis), which we attribute to the onset of dynamic subsidence at this time (Gurnis, 1992) (Fig. 6a). Therefore, the paleogeographic modifications of the southwestern Pangea margin are compatible with the tomotectonic reconstructions and igneous records that indicate a Permian-Triassic slab loss stage followed by a progressive reactivation of subduction in the latest Triassic-Early Jurassic.

We introduced a new paragraph discussing this issue in new lines 428-458 and Fig. 6.

References:

Limarino, C. O., & Spalletti, L. A. (2006). Paleogeography of the upper Paleozoic basins of southern South America: An overview. *Journal of South American Earth Sciences*, 22(3-4), 134-155.

Limarino, C. O., Césari, S. N., Spalletti, L. A., Taboada, A. C., Isbell, J. L., Geuna, S., & Gulbranson, E. L. (2014). A paleoclimatic review of southern South America during the late Paleozoic: a record from icehouse to extreme greenhouse conditions. *Gondwana Research*, 25(4), 1396-1421.

Ford, D., & Golonka, J. (2003). Phanerozoic paleogeography, paleoenvironment and lithofacies maps of the circum-Atlantic margins. *Marine and petroleum geology*, 20(3-4), 249-285.

Torsvik, T. H., & Cocks, L. R. M. (2013). Gondwana from top to base in space and time. *Gondwana Research*, 24(3-4), 999-1030.

Vicente, J.-C., 2005. Dynamic paleogeography of the Jurassic Andean Basin: pattern of transgression and localisation of main straits through the magmatic arc. *Revista de la Asociación Geológica Argentina*, v. 60, pp. 221–250

Reviewer #1: Regarding the color scale for the vote map (figure 2d): I commented previously that the colormap should be changed, and the authors have adopted a different colormap, avoiding the use of a diverging colorscale. However, although the revision uses a new color map, the alternative version is still diverging and so suffers from exactly the same problem as the previous one. To be specific, an example of an appropriate color map would be one that uses a bright color for the highest value (26?) and gradually becomes less bright continuously towards zero (note that the difference between 0 votes and 1 vote is no more or less significant than the difference between 1 and 2 votes, yet the colormap used here emphasises the region with few (but not zero) votes in bright red, and the boundary between 1 and 0, despite the fact that this boundary has no special meaning.

Gianni and Navarrete: The reviewer is right. We have modified the colormap with a palette that does not emphasize low vote regions.

Reviewer #2 (Remarks to the Author):

Dear editor,

Hereby I send you my review of Guido M. Gianni and César R. Navarrete's paper 'A catastrophic slab loss in Pangea preserved in the lower mantle structure and the igneous record', resubmitted

for publication in Nature Communications.

The authors took the previous comments seriously and corrected the necessary parts as requested. They have taken most of the corrections of reviewer #1 and 3, also they tried to reply all their questions in detail. The answers that are given for the comments are satisfactory. Figures are clearer and better in general terms.

This paper shows us that the deeper tomographic anomalies can still be linked to their surface manifestations, and it brings another constraint to the plate reconstructions for the early stages of Phanerozoic. With few minor corrections/additions I recommend this paper to be published.

Reviewer #2: In this paper, it is still not clear where the anomaly of Choiyoi magmatic province, slab gap, is located in present-day plate configuration. It can be deduced from Fig. S2 but as I mentioned earlier, a small text about the displacement between the present-day location of the Choiyoi Magmatic Province and the slab gap can also give insight to the reader who is interested in the relative motion between the lithosphere and the average mantle.

Gianni and Navarrete: The reviewer is right. We added a small sentence in new lines 156-158 indicating that both anomalies are located between 1950 and 2350 km from the present Andean trench. Also, we plotted the current position of South America in Fig. 2d to depict more clearly the relative continental drift respect to these slab relics.

Reviewer #2: Misspellings in Fig. 5, "North" Patagonia Terrane

Gianni and Navarrete: This was corrected as suggested.

Reviewer #2: Line 320: How close is it?

Gianni and Navarrete: We have introduced new lines 322-325 describing in more detail the basin shape and size as suggested.

Reviewer #2: The use of early/late should be corrected in the main and supplementary text according to the following rule (Line 144, 389 etc.). Time-rock (chronostratigraphic) units (lower/upper) should be used for attribution of age to rocks, formations, biostratigraphic zones, unconformities, and seismic reflectors and, consequently, on well logs, columnar and seismic sections, cross-sections, and stratigraphic correlation diagrams. Time-rock units have a base, a top, and a thickness.

Time (geochronological) units (early/late) should be used for describing the time of occurrence of historical events such as periods of erosion, transgression, folding, faulting, faunal extinction, and oil generation

Gianni and Navarrete: The reviewer is right. We have corrected the manuscript and the supplementary file following this suggestion.

Reviewer #3 (Remarks to the Author):

Overall the authors have done a great job in replying to the reviewer's comments and modifying the paper, which is now much stronger and worthy of publishing. The revised manuscript does a very nice job of relating the issues, explaining the methodology and is now an important contribution not only on the origin of the Choiyoi province, but also on combining seismic tomography, geochemical data and geologic observations. The resulting manuscript is a step forward on mechanisms that can create a major silicic igneous province (SLIP), supports an important break in subduction along a section of the margin of the Gondwana supercontinent and opens the door for future discussions.

A few minor comments on the age range of the Choiyoi province and figures are below. The paper might also benefit from being edited one more time for English expression as there are several places where it could be improved. Note also that the number notations to the references for the geochemical data in the supplementary data need to be updated as the numbers are from the initially submitted paper, not the revised version.

Gianni and Navarrete: Both English and references have been revised as suggested.

A few specific comments keyed to line number.

Reviewer #3: Line 6. In the abstract, the Choiyoi province is given a specific age between 286 and 247 Ma. As the term "Choiyoi province" has been used loosely in terms of a time range, the range of ages is important as the limits have implications as to how much of the early plutonic and later silicic magmatism is included. In contrast to the abstract, line 30 of the paper says ca. 286 and 247 Ma and the discussion section later in the paper calls for diachronies in events. A short discussion on the time frame could add clarity.

Gianni and Navarrete: The reviewer is right. To avoid confusion we simply refer to mid-Permian-Triassic magmatism in the abstract. Also, we introduced a sentence clarifying this issue in new lines 35-37.

Reviewer #3: Line 111. Just a thought, but what is the relation of this to the old polar wandering curves discussed in relation to the Choiyoi province by Kay et al. (1989) and Ramos and Kay

(1991)?

Gianni and Navarrete: This is a good point. Wall-like geometries in the Georgia and Sao Francisco slabs are indeed quite compatible with the early polar wander path for Pangea of Valencio et al. (1983) for this period. Both observations attest to a relatively stationary character of the Pangea margin at this time.

Reviewer #3: Line 263. Is this 900 km from the current trench or a trench at the time of the Choiyoi event? Please clarify.

Gianni and Navarrete: The reviewer is right. The 900 km mentioned in the text would be from the current trench, and aprox. 1000 km would be the outcrop distance from the paleo-trench. We have clarified this issue in new lines 265-266.

Reviewer #3: Figure 1. Indicate in caption that when first mentioned that GI is Georgia Islands and SF is Sao Francisco or better yet write out the names on the figure.

Gianni and Navarrete: The meaning of GI and SF abbreviations is indicated at the end of the caption.

Reviewer #3: Figure 3. Outline for outcrop distance from current trench is not clearly distinct from that from that from the paleo-trench making figure somewhat confusing.

Gianni and Navarrete: We have corrected this figure as suggested.

Reviewer #3: In figure 4, it might be more logical to have the III-IV cross-section, which is further south and more complex, as the third part of the figure and the I-II section, which is further north in the middle of the figure. Exchange labels III and IV with I and II.

Gianni and Navarrete: We have corrected this figure as suggested.

RESPONSE TO REVIEWER COMMENTS: In general, Gianni and Navarrete did an impressive job in responding to the reviewer's comments.

I did find one strange comment in reference to my own papers in the author's response to the comment of Reviewer 1 on the Nelson and Cottle (2019) paper. In section 1 of their reply, I was

surprised to see Kay et al. (1989) and Mpodozis and Kay (1992) included among the papers calling for an uninterrupted subduction zone setting for the Choiyoi province. Initially, it left me wondering what they meant by an “uninterrupted” subduction zone setting as our papers clearly called for a break in the subduction process. I concluded it was a glitch in the response as it conflicts with what is said in the text.

Gianni and Navarrete: Prof. Kay is right; it was a mistake in our response to the reviewer.

We are grateful with the three reviewers for providing valuable feedbacks to our manuscript.

After this second revision we are confident that the findings and the approach followed in our study can be useful, not only to geoscientists trying to solve the geodynamic settings of SLIPs (subduction vs. non-subduction settings) but also to those aiming to detect slab break-off events from the geological record in pre-Cenozoic times, which can also be of exploratory interest. Furthermore, as the slab break-off process modifies the upper plate tectonic regime and induces kilometric-scale surface uplift, it ultimately impacts local climate and biodiversity (e.g. Webb et al. 2017). Hence, as illustrated in our revised manuscript, the approach described in this study can be used by other geoscientists to understand paleogeographic modifications in ancient active margins.

Dr. Guido Gianni and Dr. César Navarrete

REVIEWERS' COMMENTS

Reviewer #1 (Remarks to the Author):

The authors have made a good effort to address the previous suggestions. My additional comments relate to new material that has been added to the revision. The required changes would be minor.

Two paragraphs beginning on line 428:

This is the main new material added to the revised manuscript. The main point that I think needs clarifying concerns the magnitude of uplift, and whether the magnitudes expected from geodynamic considerations are enough to support the arguments about orographic precipitation. Specifically, we are given some ballpark figures for the rates of uplift / subsidence that can be induced by various subduction zone dynamics. Just before this, we are told that topographic barriers can influence the local climate (supported by some modelling using PaleoDEMs), but we are not told how large these topographic barriers are (for example, the elevation from the model PaleoDEM used in the paleoclimate model is a known quantity, so what is it? More generally, would a 1 km high barrier be enough? 2 km?).

A stylistic point - there are many sentences in these paragraphs (including three in succession) that start with the word 'this', which I find makes the flow hard to follow.

New text starting at line 478:

The main issue I have with this section is that the new analysis is only considering seismic tomographic evidence for past subduction, not a more holistic approach (as the authors have applied to their own region of interest, considering all the geological data). However, tomography alone is unlikely to tell the whole story, especially not for slabs that would lie at mid-mantle depths in the southern hemisphere where the resolution of tomography models is degraded relative to other regions. For this reason, I find the suggestion beginning on line 486 regarding the Whitsunday SLIP to be unconvincing. Whilst the history of subduction in this part of the Gondwana margin might be the subject of debate in the Cretaceous, it is much better known further towards New Zealand (where there is abundant evidence for subduction ongoing until around 100 Ma, which is when the Hikurangi Plateau arrived in the subduction zone and jammed it up). Yet, the tomography slice depicted in figure 7 shows no sign of this subduction, illustrating that the problem lies with the tomography (or perhaps the method of interpreting it in terms of subduction history). Therefore, the text on lines 486 should be rewritten to reflect the ambiguity of interpretation in this region (in fact, I see no reason why the Whitsunday example could not follow the slab-breakoff model, it's just that the evidence to support either model is equivocal). (A useful recent work related to this issue is that of Tucker et al, 2020, GSA Bulletin)

Reviewer #1 (Remarks to the Author):

The authors have made a good effort to address the previous suggestions. My additional comments relate to new material that has been added to the revision. The required changes would be minor.

Two paragraphs beginning on line 428:

This is the main new material added to the revised manuscript. The main point that I think needs clarifying concerns the magnitude of uplift, and whether the magnitudes expected from geodynamic considerations are enough to support the arguments about orographic precipitation. Specifically, we are given some ballpark figures for the rates of uplift / subsidence that can be induced by various subduction zone dynamics. Just before this, we are told that topographic barriers can influence the local climate (supported by some modelling using PaleoDEMs), but we are not told how large these topographic barriers are (for example, the elevation from the model PaleoDEM used in the paleoclimate model is a known quantity, so what is it? More generally, would a 1 km high barrier be enough? 2 km?).

Gianni and Navarrete: The reviewer is correct. We now indicate in the text that the Choiyoi SLIP orographic barrier must have been ≥ 1 km, the minimum height to generate a noticeable rain shadow effect (e.g. Blisniuk et al., 2005, *EPSL*; Bucher et al. 2019, *GSA Bulletin*) such as the one recorded in the southwestern Pangea basins (see new line 438-439). Noteworthy, surface uplift magnitudes above 1 km are easily explained by post-slab break-off uplift (<2-6 km; Buitter et al., 2002, *Tectonophysics*; Duretz et al., 2011, *Tectonophysics*) (see lines 445-448). PaleoDEM included heights of ~1-1.5 km for the Permian in the area associated with the Choiyoi high.

A stylistic point - there are many sentences in these paragraphs (including three in succession) that start with the word 'this', which I find makes the flow hard to follow.

Gianni and Navarrete: The reviewer is right. We have corrected the paragraph as suggested.

New text starting at line 478:

The main issue I have with this section is that the new analysis is only considering seismic tomographic evidence for past subduction, not a more holistic approach (as the authors have applied to their own region of interest, considering all the geological data). However, tomography alone is unlikely to tell the whole story, especially not for slabs that would lie at mid-mantle depths in the southern hemisphere where the resolution of tomography models is degraded relative to other regions.

For this reason, I find the suggestion beginning on line 486 regarding the Whitsunday SLIP to be unconvincing. Whilst the history of subduction in this part of the Gondwana margin might be the subject of debate in the Cretaceous, it is much better known further towards New Zealand (where there is abundant evidence for subduction ongoing until around 100 Ma, which is when the Hikurangi Plateau arrived in the subduction zone and jammed it up). Yet, the tomography slice depicted in figure 7 shows no sign of this subduction, illustrating that the problem lies with the tomography (or perhaps the method of interpreting it in terms of subduction history). Therefore, the text on lines 486 should be rewritten to reflect the ambiguity of interpretation in this region (in fact, I see no reason why the Whitsunday example could not follow the slab-breakoff model, its just that the evidence to support either model is equivocal).

(A useful recent work related to this issue is that of Tucker et al, 2020, GSA Bulletin).

Gianni and Navarrete: The reviewer is right. We have modified this sentence warning about the limitations of seismic tomography models at mid-mantle depths at these latitudes as suggested by reviewer 1 (see new line 492-494). Also, please note that we did suggest a slab-break-off origin for the Whitsunday SLIP in following lines 512-514.

We are grateful to reviewer 1 for the constructive comments throughout the revision of our manuscript.

Guido M. Gianni and César R. Navarrete